# DatasetDM: Synthesizing Data with Perception Annotations Using Diffusion Models

**Weijia Wu**[1,3]    **Yuzhong Zhao**[2]    **Hao Chen**[1]    **Yuchao Gu**[3]    **Rui Zhao**[3]    **Yefei He**[1]
**Hong Zhou**[1*]    **Mike Zheng Shou**[3*]    **Chunhua Shen**[1,4]

[1]Zhejiang University, China    [2]University of Chinese Academy of Sciences, China
[3]Show Lab, National University of Singapore    [4]Ant Group

## Abstract

Current deep networks are very data-hungry and benefit from training on large-scale datasets, which are often time-consuming to collect and annotate. By contrast, synthetic data can be generated infinitely using generative models such as DALL-E and diffusion models, with minimal effort and cost. In this paper, we present DatasetDM, a generic dataset generation model that can produce diverse synthetic images and the corresponding high-quality perception annotations (*e.g.*, segmentation masks, and depth). Our method builds upon the pre-trained diffusion model and extends text-guided image synthesis to perception data generation. We show that the rich latent code of the diffusion model can be effectively decoded as accurate perception annotations using a decoder module. Training the decoder only needs less than $1\%$ (around 100 images) manually labeled images, enabling the generation of an infinitely large annotated dataset. Then these synthetic data can be used for training various perception models for downstream tasks.

To showcase the power of the proposed approach, we generate datasets with rich dense pixel-wise labels for a wide range of downstream tasks, including semantic segmentation, instance segmentation, and depth estimation. Notably, it achieves (1) state-of-the-art results on semantic segmentation and instance segmentation; (2) significantly more robust on domain generalization than using the real data alone; and state-of-the-art results in zero-shot segmentation setting; and (3) flexibility for efficient application and novel task composition (*e.g.*, image editing).

The project website is at: weijiawu.github.io/DatasetDM.

## 1    Introduction

Modern deep-learning models for perception tasks often require a large amount of labeled data to achieve good performance. Unfortunately, collecting large-scale data and labeling the corresponding pixel-level annotations is a time-consuming and expensive process. For example, collecting images of urban driving scenes requires physical car infrastructure, and labeling a segmentation annotation for a single urban image in Cityscapes [10] can take up to 60 minutes. Moreover, in certain specialized domains, such as medical or human facial data, collecting relevant information can be challenging or even impossible, owing to privacy concerns or other factors. The above challenges can be a barrier to advancing artificial intelligence in computer vision.

To reduce costs, many previous researchers have primarily focused on weakly supervised [57] and unsupervised solutions [51] to address the problem. For instance, certain segmentation priors [1, 2, 32] use weak or inexpensive labels to train robust segmentation models. With the advancement of

---

*H. Zhou and M. Shou are the corresponding authors.

37th Conference on Neural Information Processing Systems (NeurIPS 2023).

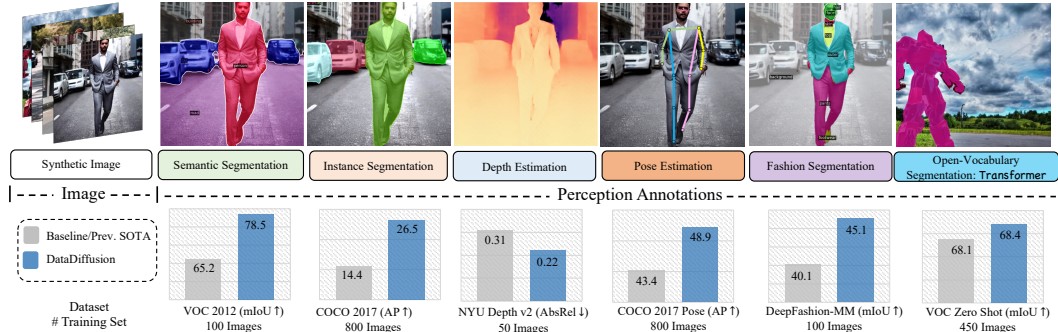

Figure 1: **Synthetic Data from DatasetDM**. The high-quality and infinitely numerous synthetic images with perception annotation can yield significant improvements for various downstream tasks.

generative models, such as DALL-E [40], Stable Diffusion [41], some researchers have begun to explore the potential of synthetic data, attempting to use it to assist in training models, or even replace real data for perception task. Most works focus on classification, face recognition [34, 49, 22], salient object detection [54] and segmentation tasks [31, 53, 58], with only a minority trying to address problems such as human pose estimation [18] or medical image analysis [21]. In the era of GANs, DatasetGAN [58] and BigDatasetGAN [29] are recognized as pioneering works that utilize the feature space of pre-trained GANs and design a shallow decoder for generating pixel-level labels in the context of segmentation tasks. Following the two works, Hands-off [55] extends this approach to multitasking scenarios, such as depth estimation. However, these methods still suffer from three major drawbacks: 1) Due to the limitations of the representation ability of early (up to 2020) GAN models, the quality of the synthesized data is often dissatisfactory, leading to an inferior performance on downstream tasks. 2) These methods primarily focus on independent downstream tasks and no one tries to explore a unified data synthesis paradigm with a generalized decoding framework. 3) The training cost is still relatively high, while these methods do not make full use of the visual knowledge contained within the latent codes of powerful text-to-image models.

By leveraging large-scale datasets of image-text pairs (*e.g.*, LAION5B [45]), recent text-to-image diffusion models (*e.g.*, Stable Diffusion [41]) present phenomenal power in generating diverse and high-fidelity images with rich texture, diverse content, and reasonable structures. The phenomenon suggests that large text-to-image diffusion models can *implicitly* learn valuable, and rich high-level and low-level visual representations from massive image-text pairs. It is natural ask: *Can we leverage the knowledge learned by these models to generate perception annotations and extend the paradigm of text-to-image generation to text-to-data generation?*

In this paper, built upon the powerful text-to-image diffusion model, we present DatasetDM, a generalized dataset generation model that can produce an unlimited number of synthetic images and the corresponding perception annotations, as shown in Fig. 1. The key to our approach is a unified perception decoder, namely P-Decoder, that decodes the latent code of the diffusion model to perception annotations. Inspired by align/instruct tuning from LLM [50], a method for inducing output following capability with minimal human-labeled, we only use less than **1**% manually labeled images to train the decoder, enabling infinite annotated data generation. The generated data can subsequently be utilized to train any perception models for various downstream tasks, including but not limited to segmentation, depth estimation, and pose estimation. To maintain the robust image generation capabilities of the diffusion model, we freeze the model weights and use image inversion to extract the latent code, *i.e.*, multi-scale features, which are then fed into the unified P-Decoder. The P-Decoder accommodates various downstream tasks within a unified transformer framework, with only minor variations as depicted in Fig. 3.

To summarize, our contributions are four-fold:

- We introduce DatasetDM: a versatile dataset generation model featuring a perception decoder capable of producing an unlimited quantity of high-fidelity synthetic images, along with various perception annotations, including depth, segmentation, and human pose estimation.
- Visual align/instruct tuning, a method for inducing output following capability with minimal human-labeled data. With less than 1% of the original dataset, *i.e.*, around 100 images,

DatasetDM pushes the limits of text-to-image generation, pioneering a novel paradigm: *text-to-data generation*. This breakthrough is made possible by leveraging the rich latent code of the pre-trained diffusion model.

- Experiments demonstrate that the existing perception models trained on synthetic data generated by DatasetDM exhibit outstanding performance on **six** datasets across **five** different downstream tasks. For instance, the synthetic data delivers remarkable gains of 13.3% mIoU and 12.1% AP for semantic segmentation on VOC 2012 and instance segmentation on COCO 2017, respectively.

- Text-guided data generation allows for the generation of a diverse range of data, which has been shown to provide a more robust solution for domain generalization and long-tail data distribution. Moreover, DatasetDM offers a flexible approach for novel task composition, as exemplified by its ability to facilitate image editing (see Fig. 4).

## 2   Related Work

**Text-to-Image Generation.** Several mainstream methods exist for the task, including Generative Adversarial Networks (GANs)[19], Variational Autoencoders (VAEs)[28], flow-based models [12], and Diffusion Probabilistic Models (DPMs) [47, 41, 25, 20]. Recently, as a likelihood-based method, diffusion models have gained significant attention with promising generation abilities. They match the underlying data distribution by learning to reverse a noising process. Thanks to the high-quality image synthesis and the stability of training, diffusion models are quickly emerging as a new frontier [41, 40, 26, 43] in the field of image synthesis.

Text-guided diffusion models are used for text-conditional image generation, where a text prompt $\mathcal{P}$ guides the generation of content-related images $\mathcal{I}$ from a random Gaussian noise $z$. Visual and textual embeddings are typically fused using cross-attention. Recent large text-to-image diffusion models, such as Stable Diffusion [41] of Stability AI, DALL-E2 [40] of OpenAI, and Imagen [43] of Google, have shown powerful performance in generating diverse and high-fidelity images with rich textures and reasonable structures. Their impressive synthesis capabilities suggest that these models can implicitly learn valuable representations with the different semantic granularity from large-scale image-text pairs. In this paper, we leverage these learned representations (latent codes), to extend the paradigm of text-to-image generation to text-to-data generation.

**Synthetic Datasets for Perception Task.** Previous studies in dataset synthesis, such as Virtual KITTI [17] and Virtual KITTI 2 [7], primarily rely on 3D computer simulations to address standard 2D computer vision problems, such as object detection [38], scene understanding [44], and optical flow estimation [6]. However, these methods are limited by the domain of 3D models and cannot be generalized to arbitrary scenes and open-set categories. For instance, Virtual KITTI is exclusively focused on autonomous driving scenes and supports only 20 commonly occurring categories, which cannot be extended to open-scene domains like the COCO benchmark [33].

In contrast, synthetic data generated using generation models (*i.e.*, GAN [19, 35, 55] and Diffusion Model [47]) are more flexible and can support a wider range of tasks and open-world scenes for various downstream tasks, such as classification task [22], face recognition [22], salient object detection [54], semantic segmentation [29, 58, 3, 60], and human pose [18]. Inspired by the success of large-scale generative models, such as Stable Diffusion [41], trained on massive datasets like LAION5B [45], recent studies have begun to explore the potential of powerful pre-trained diffusion generative models. Based on DDPM [25], DatasetDDPM [3] design several CNN layers as the annotation decoder to generate semantic segmentation data by performing a multi-class task. Li *et al.* [31] utilized Stable Diffusion and Mask R-CNN pre-trained on the COCO dataset [33] to design and train a grounding module for generating images and semantic segmentation masks. DiffuMask [53] produces synthetic image and semantic mask annotation by exploiting the potential of the cross-attention map between text and image from the text-supervised pre-trained Stable Diffusion model. The above methods focus on semantic segmentation, which cannot handle tasks such as instance segmentation. In this paper, we take a further step by utilizing a generalized perception decoder to parse the rich latent space of the pre-trained diffusion model, enabling the generation of perception for a variety of downstream tasks, including depth, segmentation, and human pose.

**Diffusion Model for Perception Task.** Some recent works [59, 56, 52] has also attempted to directly employ diffusion models for perceptual tasks. VPD [59] explores the direct application of pre-trained

Stable Diffusion to design perception models. ODISE [56] unify pre-trained text-image diffusion and discriminative models to perform open-vocabulary panoptic segmentation. Different from these approaches, our focus lies in synthetic data augmentation for perception tasks, and design a unified transformer-based decoder to enhance more perception tasks from data aspect.

# 3 Methodology

## 3.1 Formulation

Given a language prompt $\mathcal{S}$, text-guided diffusion models generate content-related images $\mathcal{I} \in \mathcal{R}^{H \times W \times 3}$ from a random Gaussian noise $\boldsymbol{z} \sim \mathcal{N}(\mathbf{0}, \mathbf{I})$. The standard text-guided image denoising processing can be formulated as: $\mathcal{I} = \Phi_{\text{T2I}}(\boldsymbol{z}, \mathcal{S})$, where $\Phi_{\text{T2I}}(\cdot)$ refers to a pre-trained text-to-image diffusion model. In this paper, we adopt Stable Diffusion [41] as the base for the diffusion model, which consists of three components: a text encoder $\tau_\theta(\cdot)$ for embedding prompt $\mathcal{S}$; a pre-trained variational autoencoder (VAE) [14] that encodes $\mathcal{E}(\cdot)$ and decodes $\mathcal{D}(\cdot)$ latent code of images; and a time-conditional UNet ($\epsilon_\theta(\cdot)$) [42] that gradually denoises the latent vectors. To fuse visual and textual embeddings, cross-attention layers are typically used in the UNet for each denoising step. The denoising process is modeled as a Markov chain: $\boldsymbol{x}_{t-1} = f(\boldsymbol{x}_t, \epsilon_\theta)$, where $\boldsymbol{x}_t$ denote latent code at timestep $t$, and $t \in [1, T]$. The latent noise at the final timestep $T$, denoted as $\boldsymbol{x}_T$, is equivalent to the random Gaussian noise $\boldsymbol{z}$. $f(\cdot)$ is the denoising function [25].

In this paper, we design a perception decoder that can effectively parse the latent space of the UNet $\epsilon_\theta(\boldsymbol{x}_t, t, \tau_\theta(\mathcal{S}))$. By doing so, we extend the *text-to-image* generation approach to a *text-to-data* paradigm:

$$\{\mathcal{I}, \mathcal{P}_{1:k}\} = \Phi_{\text{T2D}}(\boldsymbol{z}, \mathcal{S}), \tag{1}$$

where $\mathcal{P}_{1:k}$ denotes the corresponding perception annotations, and $k$ is the number of the supported downstream tasks. In fact, the paradigm can support any image-level perception task, such as semantic segmentation, instance segmentation, pose estimation, and depth estimation.

## 3.2 Method Overview

This paper introduces a novel paradigm called *text-to-data generation*, which extends text-guided diffusion models trained on large-scale image-text pairs. Our key insight is that using a small amount of real data (using less than $1\%$ existing labeled dataset) and a generic perception decoder to interpret the diffusion latent spaces, results in the generation of infinite and diverse annotated data. Then the synthetic data can be used to train any existing perception methods and apply them to real images.

The proposed DatasetDM framework, presented in Fig. 2, comprises two stages. 1) The first stage—**Training**—involves using diffusion inversion (§3.3) to obtain the latent code of the real image and extract the text-image representations (§3.3). These representations and their corresponding annotations are then used to train the perception decoder (§3.4). 2) The second stage—**Inference** (§3.5)—uses GPT-4 to enhance the diversity of data and generates abundant images, while the P-Decoder produces corresponding perception annotations such as masks and depth maps.

## 3.3 Hypercolumn Representation Extraction

The first step in the training stage of DatasetDM is to extract the hypercolumn representation of real images from the latent space of the diffusion model, as shown in Fig. 2(a). To achieve this, we employ the diffusion inversion technique, which involves adding a certain level of Gaussian noise to the real image and then extracting the features from the UNet during the denoising process.

**Image Inversion for Diffusion Model.** Give a real image $\mathcal{X} \in \mathcal{R}^{H \times W \times 3}$ from the training set, the diffusion inversion (diffusion forward process) is a process that approximates the posterior distribution $q(\boldsymbol{x}_{1:T}|\boldsymbol{x}_0)$, where $\boldsymbol{x}_0 = \mathcal{E}(\mathcal{X})$. This process is not trainable and is based on a fixed Markov chain that gradually adds Gaussian noise to the image, following a pre-defined noise schedule $\beta_1, \ldots, \beta_T$ [25]:

$$q(\boldsymbol{x}_t|\boldsymbol{x}_{t-1}) := \mathcal{N}(\boldsymbol{x}_t; \sqrt{1 - \beta_t}\boldsymbol{x}_{t-1}, \beta_t\mathbf{I}), \tag{2}$$

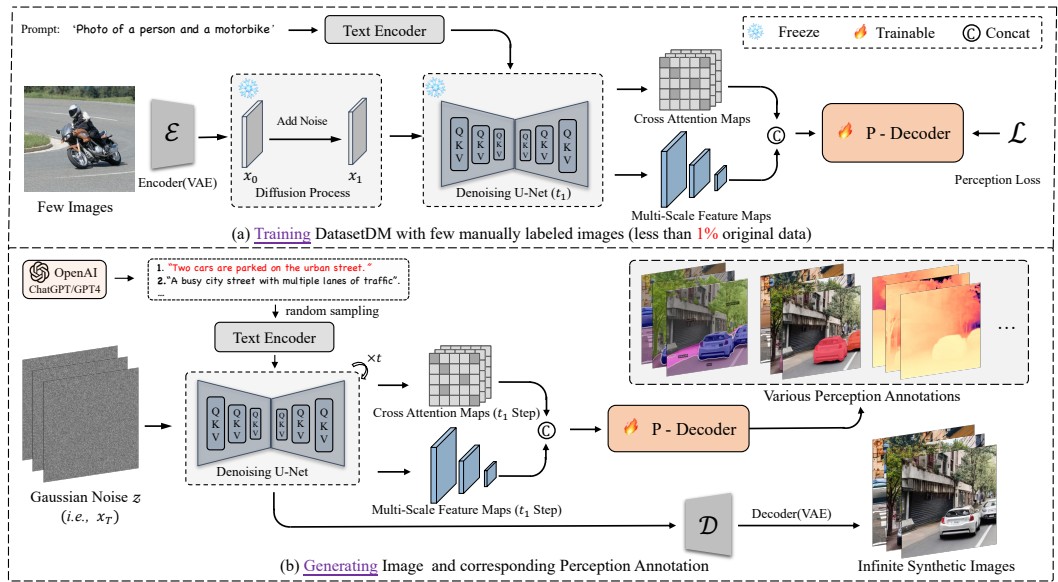

Figure 2: **The overall framework of DatasetDM.** DatasetDM consists of two main steps: 1) Training. Using diffusion inversion to extract the latent code from a small amount of data and then train the perception decoder. 2) Text-guided data generation. A large language model such as GPT-4 is utilized to prompt infinite and diverse data generation for various downstream tasks.

where $t$ represents the $t$-th time step, and we set it to 1 for a single forward pass in our paper. A single forward pass for visual representation extraction usually provides two advantages, *i.e.*, faster convergence and better performance [56].

**Text-Image Representation Fusion.** With the latent code $x_t$ of the real image and the corresponding language prompt $\mathcal{S}$, we extract the multi-scale feature maps and cross attention maps from the UNet $\epsilon_\theta$ as follows:

$$\{\mathcal{F}, \mathcal{A}\} = \epsilon_\theta(x_t, t, \tau_\theta(\mathcal{S})), \tag{3}$$

where $\mathcal{S}$ for training set is simply defined using a template "a photo of a [CLS]$_1$, [CLS]$_2$, ...". During the data generation phase, GPT-4 is used to provide diverse prompt languages. The variable $\mathcal{F}$ denotes the multi-scale feature maps from four layers of the U-Net, corresponding to four different resolutions, *i.e.*, $8 \times 8$, $16 \times 16$, $32 \times 32$, and $64 \times 64$, as illustrated in Fig. 2. Additionally, $\mathcal{A}$ represents the cross-attention maps of text-to-image from the 16 cross-attention layers in the U-Net, which implement the function $\mathcal{A} = \text{softmax}\left(\frac{QK^T}{\sqrt{d}}\right)$, where $d$ is the latent projection dimension. We group the 16 cross-attention maps into 4 groups with the same resolutions, and compute their average within each group, which results in the average cross-attention maps $\hat{\mathcal{A}}$.

Prior works [53, 59, 24] have proved the effectiveness of class-discriminative and localization of cross-attention map between the visual embedding and the conditioning text features. Thus we concatenate the cross-attention maps $\hat{\mathcal{A}}$ and the multi-scale feature maps $\mathcal{F}$ to obtain the final extracted hyper-column representation, and further use a $1 \times 1$ convolution to fuse them: $\hat{\mathcal{F}} = \text{Conv}([\mathcal{F}, \hat{\mathcal{A}}])$.

### 3.4 Perception Decoder

The P-Decoder is utilized to translate the representation $\hat{\mathcal{F}}$ into perception annotations, which are not limited to a specific type for each downstream task. To achieve this goal, inspired by previous works [8, 61], we devised a generalized architecture. This architecture is depicted in Fig. 3, with only *minor variations* (*i.e.*, whether to startup some layers) for each downstream task. For example, the pixel decoder and transformer decoder are required for generic segmentation, while only the pixel decoder is necessary for depth and pose estimation.

**Generic Image Segmentation.** In Fig. 3-(a), we present the adaptation for semantic and instance segmentation tasks, which includes two components: the pixel decoder and the transformer decoder.

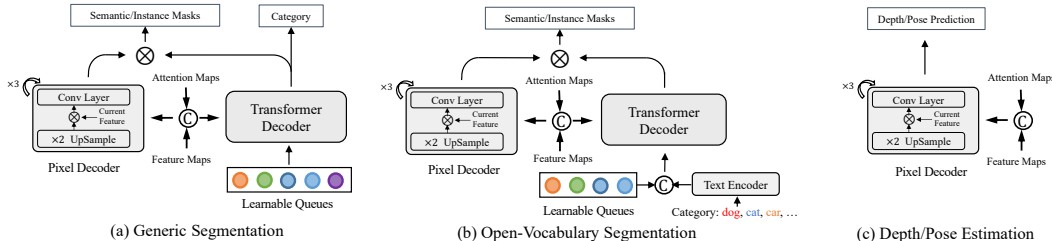

Figure 3: **Various types of tasks with our proposed P-Decoder.** The proposed decoder is a generalized architecture for the six supported tasks, with only minor variations required for different downstream applications, *i.e.*, determining whether to activate certain layers.

Similar to Mask2former [8], the transformer decoder comprises a stack of transformer layers with cross-attention and self-attention. This component refines the queries and renders the outputs. The pixel decoder is made up of multiple upsampling CNN layers, and it is used to obtain per-pixel embeddings. Given the representation $\hat{\mathcal{F}}$ and $N$ learnable queues $\{Q_0, Q_1...Q_T\}$ as input, the decoder outputs $N$ binary masks $\mathbf{O} = o_1, \cdots, o_N \in \{0, 1\}^{N \times H \times W}$, along with the corresponding category. This is achieved through simple matrix multiplication between the outputs of the transformer decoder and the pixel decoder. Following Mask2former [8], one query is responsible for a class or instance for semantic and instance segmentation, respectively. To optimize the mask prediction, we use the binary cross-entropy loss [9] and dice loss [37]. On the other hand, the cross-entropy loss is used for classification prediction.

**Open-Vocabulary Segmentation.** Based on the architecture of generic segmentation, a semantics-related query concept is proposed to support the open-vocabulary segmentation task, as shown in Fig 3-(b). During the training stage, given the category name $\{C_0, C_1...C_K\}$ (*e.g.*, dog, cat) on an image from the training set, we use the text encoder $\tau_\theta(\cdot)$, *e.g.*, CLIP [39] to encode them into the embedding space. Then concatenating them with the queries to equip the semantics of class vocabulary to the query as follows:

$$\hat{Q}_i = \text{MLP}([Q_i, \tau_\theta(C_j)]), \tag{4}$$

where $Q_i$ and $C_j$ is the $i$-th query embedding and $j$-th class. MLP refers to a learned MLP, used to fuse the class embedding and learnable query embedding. Thus DatasetDM can generate an open-vocabulary mask by incorporating a new class name, as illustrated in Fig. 5 (b).

**Depth and Pose Estimation.** For depth and pose estimation, the output format is predetermined, eliminating the need to differentiate between classes or instances. In this context, the pixel decoder is only required to predict a fixed number of maps $\mathbf{O} \in \mathcal{R}^{M \times H \times W}$. The value of $M$ is set to either 1 or 17 (corresponding to 17 human key points), depending on whether the task is depth or human pose estimation. For human pose estimation, we use the mean squared error as the loss function, and the ground truth heatmaps are generated by applying a 2D Gaussian with a standard deviation of 1 pixel centered on each key point. As for depth estimation, we update the loss function from the classic scale-invariant error [30, 13].

### 3.5 Prompting Text-Guided Data Generation

In Fig. 2 (b), we present the inference pipeline for text-guided data generation. There are two main differences compared to the training phase: firstly, the prompts come from a large language model instead of a fixed template, and secondly, the denoising process is extended to $T$ steps to obtain the synthetic images. Large language model, *i.e.*, GPT-4, is adopted to enhance the diversity of generative data, while recent works [16, 23, 4] have proven their powerful understanding and adaptability for the real world. As shown in Fig. 4, instead of the template-based prompts from humans, we guide GPT-4 to produce diverse, and infinite prompts. For different downstream tasks and datasets, we give different guided prompts for GPT-4. For example, as for the urban scene of Cityscapes [10], the simple guided prompt is like 'Please provide 100 language descriptions of urban driving scenes for the Cityscapes benchmark, containing a minimum of 15 words each. These descriptions will serve as a guide for Stable Diffusion in generating images.' In this approach, we collected $L$

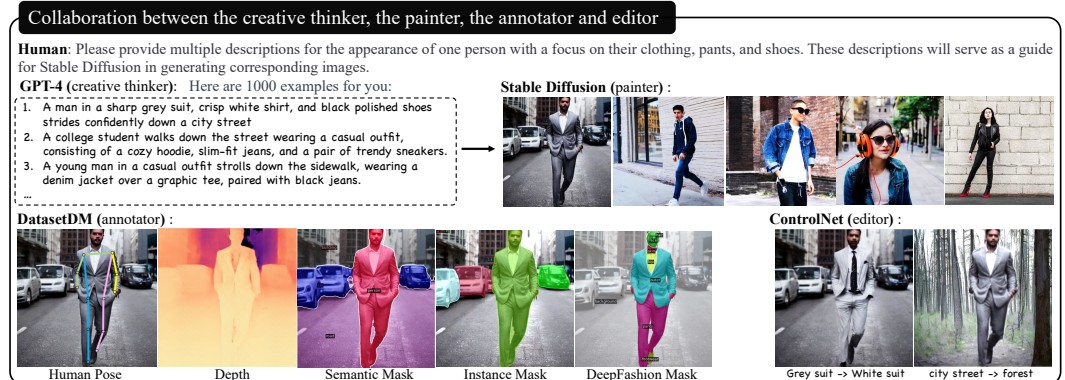

Figure 4: **Collaboration between GPT-4 and Diffusion Model.** Large language models, *e.g.*, GPT-4, can provide diverse and unrestricted prompts, enhancing the diversity of generated images.

| Zero-Shot Segmentation | | Long-Tail Segmentation | |
|---|---|---|---|
| Seen Class | Unseen Class | Head Class | Tail Class |
| aeroplane (0), bicycle (1), bird (2), boat (3), bottle (4), bus (5), car (6), cat (7), chair (8), cow (9) , diningtable (10), dog (11), horse (12), motorbike (13), person (14) | potted plant (15), sheep (16), sofa (17), train (18), tvmonitor (19) | aeroplane (0), bicycle (1), bird (2), boat (3), bottle (4), bus (5), car (6), cat (7), chair (8), cow (9) | diningtable (10), dog (11), horse (12), motorbike (13), person (14), potted plant (15), sheep (16), sofa (17), train (18), tvmonitor (19) |

Table 1: **Details for Zero-Shot and Long-tail Segmentation on VOC 2012 [15].**

text prompts, which average around 100 prompts for each dataset. For each inference, a random prompt is sampled from this set.

## 4 Experiments

### 4.1 Implementation details

**Architecture and Training Details.** Stable diffusion V1 [41] model pre-trained on the LAION5B [45] dataset is used as our text-to-image diffusion model. The decoder architecture of Mask2Former [8] was selected as the base architecture for our P-Decoder. And we use 100 queries for the segmentation task. For all tasks, we train DatasetDM for around $50k$ iterations with images of size $512 \times 512$, which only need one Tesla V100 GPU, and lasted for approximately 20 hours. Optimizer [36] with a learning rate of 0.0001 is used.

**Downstream Tasks Evaluation.** To comprehensively evaluate the generative image of DatasetDM, we conduct seven groups of experiments for the supported six downstream tasks. *Semantic Segmentation.* Pascal-VOC 2012 [15] (20 classes) and Cityscapes [11] (19 classes), as two classical benchmark are used to evaluate. We synthesized $2k$ images for each class in both datasets, resulting in a total of $40k$ and $38k$ synthetic images for Pascal-VOC 2012 and Cityscapes, respectively. The synthetic data is subsequently utilized to train Mask2former [8] and compared to its real data counterpart on a limited dataset setting (around 100 images). *Instance Segmentation.* For the COCO2017 [33] benchmark, we synthesized $1k$ images for each class, resulting in a total of $80k$ synthetic images. Similarly, Mask2former [8], as the baseline, is used to evaluate the synthetic data. We evaluate only the class-agnostic performance, where all the 80 classes are assigned the same class ID. *Depth Estimation.* We synthesized a total of $80k$ synthetic images for NYU Depth V2 [46]. And using Depthformer [30][2] to evaluate our synthetic data. *Pose Estimation.* We generated a set of $30k$ synthetic images for COCO2017 Pose dataset [33] and employed HRNet [48] as the baseline model to assess the effectiveness of our approach. *Zero-Shot Semantic Segmentation.* Following Li *et al.* [31], Pascal-VOC 2012 [15] (20 classes) is used to evaluate. We train DatasetDM with only 15 seen categories, where each category including 30 real images, and synthesized a total of $40k$ synthetic images for 20 categories. *Long-tail Semantic Segmentation.* The categories of VOC 2012 are divided into head (20 images each class) and tail classes (2 images each class). Then we train DatasetDM with these data and generate synthetic data. *Human Semantic Segmentation.* We synthesized a total

---

[2]https://github.com/zhyever/Monocular-Depth-Estimation-Toolbox

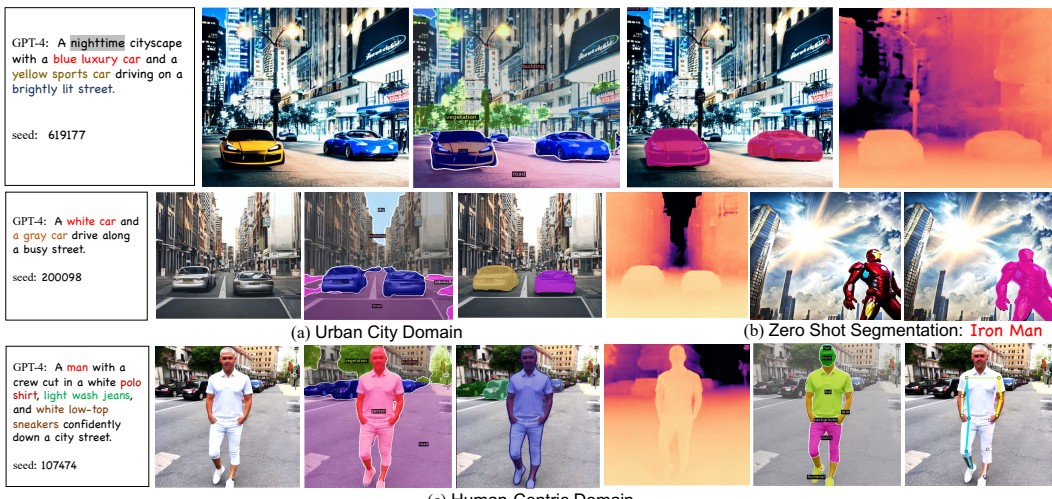

(a) Urban City Domain        (b) Zero Shot Segmentation: Iron Man

(c) Human-Centric Domain

Figure 5: **Examples of generated data for DatasetDM.** Our method can produce semantic/instance segmentation, depth, and human pose annotation across various domains.

| method | VOC (Semantic Seg.)/% | | | COCO2017 (Instance Seg.)/% | | | NYU Depth V2 (Depth Est.) | | | COCO2017 (Pose Est.)/% | | |
|---|---|---|---|---|---|---|---|---|---|---|---|---|
| | # real | # synth. | mIoU | # real | # synth. | AP | # real | # synth. | REL ↓ | # real | # synth. | AP |
| Baseline | 100 | - | 65.2 | 400 | - | 14.4 | 50 | - | 0.31 | 800 | - | 42.4 |
| **DatasetDM** | 100 | 40k | 78.5 | 400 | 80k | 26.5 | 50 | 35k | 0.21 | 800 | 80k | 47.5 |

Table 2: **Downstream Tasks.** 'real' and 'synth.' denote real and synthetic images, respectively. The backbones of baselines for four tasks are 'Swin-B', 'Swin-B', 'Swin-L', and 'HR-W32', respectively.

of $20k$ synthetic images for DeepFashion-MM [27] (24 classes). Mask2former [8] is used to evaluate the synthetic data. We split DeepFashion-MM into a set of 100 training images, and 12,500 testing images. Further details, such as prompts, can be found in the supplementary material.

**Class Split for Zero-Shot and Long-Tail Segmentation.** Table 1 provides a comprehensive overview of the class distribution for both zero-shot and long-tail scenarios. The division for zero-shot classes is consistent with previous studies [5, 53, 31]. The configuration for long-tail data distribution is firstly established in this paper.

## 4.2 Main Results

Table 2 provides a basic comparison of the selected four downstream tasks. More additional experiments can be found in Table 3 and Table 4, as well as in the Supplementary Material (*i.e.*, Pose Estimation, Depth Estimation, Zero-Shot Segmentation, Fashion Segmentation, and others).

**Semantic Segmentation.** Table 4 displays the performance on VOC 2012. Using only 100 real images (5 images per class), training with purely synthetic data from DatasetDM achieves a 73.7% mIoU, an 8.5% improvement compared to using real data alone. Moreover, when jointly training with the 100 real images, further improvement is observed, resulting in a mIoU of 78.5%. Table 5 displays the performance on Cityscapes [11]. To compare with previous methods [55], we also conducted experiments with a 9-classes division. DatasetDM demonstrates consistent advancements over the baseline or prior SOTA, achieving up to a 10% improvement in mIoU under each experimental setup.

**Instance Segmentation.** Table 3 presents three distinct training settings, encompassing variations in the backbone and the number of training images. Regardless of the chosen setting, DatasetDM consistently achieves an improvement of approximately 10%. Employing 800 training images (10 images per class) and the Swin-B backbone, DatasetDM yields a 12.1% increase in Average Precision (AP), resulting in a final AP of 26.5%.

**Depth Estimation**. Table 2 presents a concise comparison between synthetic and real data on the NYU Depth V2 dataset [46]. Detailed information (*e.g.*, backbone, other metrics) can be found in the supplementary material. When trained with 50 images, DatasetDM can achieve a 10% improvement compared to training solely with real images.

| method | backbone | # real image | # synthetic image | AP | AP$^{50}$ | AP$^{75}$ | AP$^S$ | AP$^M$ | AP$^L$ |
|---|---|---|---|---|---|---|---|---|---|
| Baseline | R50 | 400 | - | 4.4 | 9.5 | 3.5 | 1.1 | 3.3 | 12.1 |
| **DatasetDM** | R50 | - | 80k (R:400) | 12.2 | 24.3 | 10.9 | 1.6 | 11.3 | 30.9 |
| **DatasetDM** | R50 | 400 | 80k (R:400) | 14.8 | 29.7 | 13.0 | 2.3 | 15.1 | 36.0 |
| Baseline | Swin-B | 400 | - | 11.3 | 23.0 | 9.6 | 3.2 | 10.1 | 27.1 |
| **DatasetDM** | Swin-B | - | 80k (R:400) | 17.6 | 34.1 | 15.8 | 3.4 | 17.8 | 39.5 |
| **DatasetDM** | Swin-B | 400 | 80k (R:400) | 23.3 | 43.0 | 22.2 | 7.7 | 26.1 | 48.7 |
| Baseline | Swin-B | 800 | - | 14.4 | 28.8 | 12.7 | 5.6 | 15.7 | 29.2 |
| **DatasetDM** | Swin-B | 800 | 80k (R:800) | 26.5 | 46.9 | 25.8 | 7.7 | 29.8 | 53.3 |

Table 3: **Instance segmentation on COCO** `val2017`. 'R: ' denotes the real data used to train.

| method | backbone | # real image | # synthetic image | Sampled Classes for Comparison/% | | | | | mIoU |
|---|---|---|---|---|---|---|---|---|---|
| | | | | bird | cat | bus | car | dog | |
| Baseline | R50 | 100 | - | 54.8 | 53.3 | 69.3 | 66.8 | 24.2 | 43.4 |
| **DatasetDM** (ours) | R50 | - | 40k (R:100) | 84.7 | 74.4 | 86.0 | 79.2 | 63.7 | 60.3 |
| **DatasetDM** (ours) | R50 | 100 | 40k (R:100) | 81.7 | 82.3 | 87.7 | 77.9 | 69.3 | 66.1 |
| Baseline | Swin-B | 100 | - | 54.4 | 68.3 | 86.5 | 71.8 | 49.1 | 65.2 |
| **DatasetDM** (ours) | Swin-B | - | 40k (R:100) | 93.4 | 94.5 | 93.8 | 78.8 | 79.6 | 73.7 |
| **DatasetDM** (ours) | Swin-B | 100 | 100 (R:100) | 83.9 | 71.0 | 82.9 | 78.0 | 39.5 | 67.9 |
| **DatasetDM** (ours) | Swin-B | 100 | 400 (R:100) | 86.9 | 92.0 | 90.8 | 82.6 | 86.7 | 76.1 |
| **DatasetDM** (ours) | Swin-B | 100 | 40k (R:100) | 86.7 | 93.8 | 92.3 | 88.3 | 87.1 | 78.5 |
| Baseline | Swin-B | 10.6k (full) | - | 93.7 | 96.5 | 90.6 | 88.6 | 95.7 | 84.3 |
| **DatasetDM** (ours) | Swin-B | 10.6k (full) | 40k (R:100) | 93.9 | 97.6 | 91.9 | 89.4 | 96.1 | 85.4 |

Table 4: **Semantic segmentation on VOC 2012.** 'R: ' refers to the number of real data used to train.

**Human Pose Estimation.** For the human pose estimation task on the COCO2017 dataset, DatasetDM demonstrates significant improvements compared to the baseline trained on 800 real images, achieving a $5.1\%$ increase, as illustrated in Table 2. Similar to depth estimation, additional information can be found in the supplementary material.

**Zero Shot and Long-tail Segmentation.** Table 7 displays the results of experiments related to zero-shot and long-tail segmentation. Our model, DatasetDM, notably alleviates the effects of long-tail distribution by synthesizing a substantial amount of data for rare classes, leading to an improvement of up to $20\%$ in mIoU. Details for both tasks can be found in the supplementary material.

### 4.3 Ablation studies

| Method | Zero-Shot Setting | | | Long-tail Setting | | |
|---|---|---|---|---|---|---|
| | seen | unseen | harm. | head | tail | mIoU |
| Baseline(no Syn.) | 61.3 | 10.7 | 18.3 | 61.2 | 44.1 | 52.6 |
| Li *et al.* [31] | 62.8 | 50.0 | 55.7 | - | - | - |
| DiffuMask [53] | 71.4 | 65.0 | 68.1 | - | - | - |
| DatasetDM | 78.8 | 60.5 | 68.4 | 73.1 | 66.4 | 70.0 |

Table 7: **Zero Shot and Long-tail Segmentation on VOC 2012.** Zero Shot: following priors [31, 53], we train DatasetDM with only 15 seen categories, and tested for 20 categories. Long-tail Setting: the 20 categories are divided into head (10 classes, 20 images each class) and tail classes (10 classes, 2 images each class).

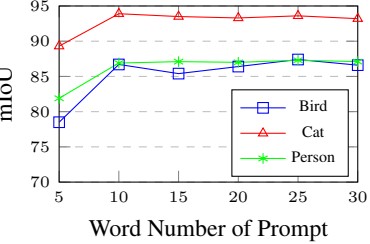

Figure 6: **Ablation of Prompt Length.**

**Diffusion Time Steps.** Table 6a depicts the influence of visual features derived from various diffusion time steps, and the maximum sampling step is 1000. We observe that a large step results in adverse outcomes, whereas the performance with a smaller step tends to remain relatively stable.

**Cross-Attention Fusion.** Fig. 6b demonstrates that the cross attention maps, $\mathcal{F}$ in step 1 can yield a modest improvement, roughly $1\%$. Interestingly, it appears that as the step size increases, the benefit becomes less pronounced. Indeed, when the step size surpasses 500, it may even result in detrimental effects. From this perspective, the utility of the cross-attention map is limited.

**Training Set Size.** Additional training data for DatasetDM can further improve synthetic data, as shown in Table 6c. The increase of the training data from 60 to 400 images precipitates the most conspicuous improvement, subsequently reaching a saturation point. With 1k training images, the

| method | backbone | # real image | # synthetic image | 8 Classes/% | | 19 Classes/% | | | |
|---|---|---|---|---|---|---|---|---|---|
| | | | | vehicle | mIoU | car | bus | bicycle | mIoU |
| Baseline | R50 | 9 | 100k+ (R:16) | 84.3 | 71.5 | 82.8 | 22.3 | 42.4 | 36.8 |
| HandsOff [55] | R101 | 16 | 100k+ (R:16) | - | 55.1 | - | - | - | - |
| HandsOff [55] | R101 | 50 | 100k+ (R:16) | - | 60.4 | - | - | - | - |
| **DatasetDM** (ours) | R50 | - | 38k (R:9) | 86.9 | 69.5 | 83.3 | 8.3 | 53.5 | 34.2 |
| **DatasetDM** (ours) | R50 | 9 | 38k (R:9) | 88.6 | 76.7 | 85.6 | 28.9 | 56.5 | 42.1 |
| **DatasetDM** (ours) | R101 | 9 | 38k (R:9) | 88.9 | 77.5 | 85.9 | 27.9 | 60.4 | 43.7 |
| Baseline | Swin-B | 9 | - | 84.1 | 74.5 | 83.3 | 27.7 | 42.0 | 41.1 |
| **DatasetDM** (ours) | Swin-B | - | 38k (R:9) | 85.7 | 73.3 | 84.3 | 20.3 | 29.1 | 37.3 |
| **DatasetDM** (ours) | Swin-B | 9 | 38k (R:9) | 89.4 | 80.0 | 87.2 | 30.2 | 66.5 | 47.4 |

Table 5: **Semantic segmentation on Cityscapes for two different split settings: 8 and 19 categories.** 'vehicle', 'car', 'bus', and 'bicycle' are sampled classes for presentation.

| step | car | dog | mIoU |
|---|---|---|---|
| 1 | 88.3 | 87.1 | 78.5 |
| 100 | 88.1 | 87.2 | 78.5 |
| 200 | 88.0 | 86.6 | 78.3 |
| 500 | 87.2 | 84.9 | 76.8 |
| 800 | 86.3 | 83.4 | 76.1 |

(a) **Visual Features $\mathcal{F}$.**

| step | car | dog | mIoU |
|---|---|---|---|
| - | 88.1 | 87.0 | 77.6 |
| 1 | 88.3 | 87.1 | 78.5 |
| 200 | 87.7 | 87.0 | 78.0 |
| 500 | 87.2 | 86.3 | 77.5 |
| 800 | 87.1 | 86.1 | 77.1 |

(b) **Cross Attention $\hat{\mathcal{A}}$.**

| # train im. | syn. | joint |
|---|---|---|
| 60 | 71.4 | 77.1 |
| 100 | 73.7 | 78.5 |
| 200 | 74.4 | 79.4 |
| 400 | 76.4 | 80.4 |
| 1,000 | 78.4 | 81.1 |

(c) **Size of Train Set.**

| prompt (# num.) | car | dog | mIoU |
|---|---|---|---|
| Human (100) | 84.9 | 84.5 | 76.6 |
| GPT-4 (100) | 85.2 | 86.1 | 77.1 |
| GPT-4 (200) | 88.0 | 86.2 | 77.3 |
| GPT-4 (500) | 88.1 | 87.1 | 78.5 |
| GPT-4 (1k) | 88.3 | 87.1 | 78.5 |

(d) **Prompt Candidates.**

Table 6: **DatasetDM Ablation on Pascal-VOC 2012 for semantic segmentation.** Swin-B is used as the backbone. 100 real images are used for (a), (b), and (d). 'Syn.' and 'Joint' denote training with only synthetic data and joint training with real data, respectively.

performance escalates to an impressive 81%, demonstrating competitive prowess for the application. Notably, 1k training images representing roughly 10% of the original data, is still relatively diminutive.

**Prompt from Language Model**. *Candidate Number.* We also investigated the impact of the number of prompt candidates, as depicted in Table 6d. With the current configurations, an increase in the number of prompts can potentially lead to a performance improvement of 2%. *Word Number of Each Prompt.* We simply study the effect of the length of prompt in Fig. 6. An increase in text length from 5 to 10 yields approximately 4% enhancement. When the text length surpasses 10, the performance appears to plateau. We argue that the upper limit is due to the current capacity of generative models.

## 5 Conclusion

In this study, we investigate using a perception decoder to parse the latent space of an advanced diffusion model, extending the text-to-image task to a new paradigm: text-guided data generation. Training the decoder requires less than 1% of existing labeled images, enabling infinite annotated data generation. Experimental results show that the existing perception models trained on synthetic data generated by DatasetDM exhibit exceptional performance across six datasets and five distinct downstream tasks. Specifically, the synthetic data yields significant improvements of 13.3% mIoU for semantic segmentation on VOC 2012 and 12.1% AP for instance segmentation on COCO 2017. Furthermore, text-guided data generation offers additional advantages, such as a more robust solution for domain generalization and enhanced image editing capabilities. We hope that this research contributes new insights and fosters the development of synthetic perception data.

## Acknowledgements

W. Wu, C. Shen's participation was supported by the National Key R&D Program of China (No. 2022ZD0118700). W. Wu, H. Zhou's participation was supported by the National Key Research and Development Program of China (No. 2022YFC3602601), and the Key Research and Development Program of Zhejiang Province of China (No. 2021C02037). M. Shou's participation was supported by the National Research Foundation, Singapore under its NRFF Award NRF-NRFF13-2021-0008, and the Ministry of Education, Singapore, under the Academic Research Fund Tier 1 (FY2022).

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
