# Supplementary Material for 'DatasetDM'

**Weijia Wu**[1,3]    **Yuzhong Zhao**[2]    **Hao Chen**[1]    **Yuchao Gu**[3]    **Rui Zhao**[3]    **Yefei He**[1]
**Hong Zhou**[1*]    **Mike Zheng Shou**[3*]    **Chunhua Shen**[1,4]

[1]Zhejiang University, China    [2]University of Chinese Academy of Sciences, China
[3]Show Lab, National University of Singapore    [4]Ant Group

## Contents

---

[*]H. Zhou and M. Shou are the corresponding authors.

37th Conference on Neural Information Processing Systems (NeurIPS 2023).

| Task | Dataset | Full Real Data | Used for DatasetDM | # synthetic image |
|---|---|---|---|---|
| Instance Segmentation | COCO 2017 [15] | 118.3k | 400 (0.3%) | 80k |
| Semantic Segmentation | VOC 2012 [8] | 10.6k | 100 (0.87%) | 40k |
| Semantic Segmentation | Cityscapes [6] | 2.9k | 9 (0.3%) | 38k |
| Semantic Segmentation | DeepFashion-MM [11] | 12.7k | 120 (0.9%) | 38k |
| Zero-Shot Segmentation | VOC 2012 [8] | 10.6k | 450 (3.9%) | 40k |
| Depth | NYU Depth V2 [19] | 24.2k | 50 (0.2%) | 35k |
| Human Pose | COCO 2017-Pose [15] | 118.3k | 800 (0.6%) | 80k |

Table 1: **Comparison of data size.** With less than 1% manually labeled images, DatasetDM can enable the generation of an infinitely large annotated dataset.

| method | backbone | input size | # real im. | # synthetic im. | AP | $AP^{50}$ | $AP^{75}$ | $AP^M$ | $AP^L$ | AR |
|---|---|---|---|---|---|---|---|---|---|---|
| Baseline | R50 | $256 \times 192$ | 800 | - | 31.3 | 62.0 | 27.7 | 30.7 | 32.0 | 36.2 |
| **DatasetDM** | R50 | $256 \times 192$ | - | 80k (R:800) | 11.4 | 28.2 | 7.8 | 6.9 | 17.7 | 14.3 |
| **DatasetDM** | R50 | $256 \times 192$ | 800 | 80k (R:800) | 36.4 | 66.7 | 35.0 | 33.0 | 40.8 | 40.1 |
| Baseline | HR-W32 | $256 \times 192$ | 800 | - | 42.4 | 73.3 | 42.1 | 39.5 | 47.0 | 46.7 |
| **DatasetDM** | HR-W32 | $256 \times 192$ | - | 80k (R:800) | 13.4 | 30.9 | 9.9 | 8.0 | 21.7 | 17.7 |
| **DatasetDM** | HR-W32 | $256 \times 192$ | 800 | 80k (R:800) | 47.5 | 75.6 | 49.3 | 44.2 | 52.6 | 51.2 |
| Baseline | HR-W32 | $384 \times 288$ | 800 | - | 43.4 | 72.2 | 44.7 | 40.5 | 47.9 | 47.5 |
| **DatasetDM** | HR-W32 | $384 \times 288$ | 800 | 80k (R:800) | 48.9 | 76.7 | 51.4 | 44.6 | 55.0 | 52.4 |

Table 2: **Human Pose Estimation on COCO `val2017`.** 'R: ' refers to the number of real data used for training DatasetDM.

# 1   Implementation details

## 1.1   Dataset Details

- Pascal-VOC 2012 [8] (20 classes) is a popular dataset for semantic segmentation in computer vision. It contains thousands of annotated images featuring 20 different object classes, such as animals, vehicles, and furniture.
- Cityscapes [6] (19 classes) is a benchmark dataset for semantic urban scene, primarily focusing on semantic segmentation tasks in computer vision. It contains high-quality pixel-level annotations of 5,000 images from 50 different cities, captured at various times of the day and under diverse weather conditions. There are 2,975 images for training and 500 images for validation.
- COCO 2017 (Common Objects in Context) [15] is a widely-used benchmark dataset for computer vision tasks, such as object detection, segmentation, and human pose estimation. It contains over 200,000 labeled images with 1.5 million object instances belonging to 80 object categories
- NYU Depth V2 [19] is designed for indoor scene understanding tasks in computer vision, specifically for depth estimation task. The NYU Depth V2 dataset contains 1,449 labeled images and 407,024 unlabeled frames, captured from 464 diverse indoor scenes.
- DeepFashion-MM [11] (24 classes) is a benchmark dataset designed for the task of clothing synthesis in the field of computer vision. It consists of 24 different clothing classes.

## 1.2   Baseline for Downstream Tasks

- **Semantic/Instance Segmentation.** We use Mask2former [4] as the baseline for comparing synthetic data to real data. We use the official code[2], maintaining all network settings, loss functions, and configurations as presented in the original code. To evaluate the effectiveness of synthetic data, we establish three settings: 1) training with purely real data, 2) training with purely synthetic data, and 3) joint training with both synthetic and real data.
- **Open-Vocabulary Semantic Segmentation.** Similar to the generic semantic segmentation, we use Mask2former [4] as the baseline. We train DatasetDM on 15 seen categories and

---

[2]`https://github.com/facebookresearch/MaskFormer`

| Collapsed label (8) | Cityscapes (Fine annotations) original labels |
|---|---|
| Void | Unlabeled (0), ego vehicle (1), rectification border (2), out of ROI (3), static (4), dynamic (5), ground (6), sidewalk (8), parking (9), rail track (10) |
| Road | Road (7) |
| Construction | Building (11), wall (12), fence (13), guard rail (14), bridge (15), tunnel (16) |
| Object | pole (17), polegroup (18), traffic light (19), traffic sign (20) |
| Nature | Vegetation (21), terrain (22) |
| Sky | Sky (23) |
| Human | Person (24), rider (25) |
| Vehicle | UCar (26), truck (27), bus (28), caravan (29), trailer (30), train (31), motorcycle (32), bicycle (33), license plate (-1) |

Table 3: **Details for** 8 **and** 19 **categories on Cityscapes [6]**.

| data aug. | baseline | backbone | # real image | # synthetic image | mIoU | Improv. |
|---|---|---|---|---|---|---|
| crop | Mask2former [4] | R50 | 100 | - | 41.5 | - |
| flip, crop, color | Mask2former [4] | R50 | 100 | - | 43.4 | +1.9 |
| crop, DatasetDM | Mask2former [4] | R50 | 100 | 40k (R:100) | 65.2 | +23.7 |
| flip, crop, color, DatasetDM | Mask2former [4] | R50 | 100 | 40k (R:100) | 66.1 | +24.6 |
| crop | Mask2former [4] | Swin-B | 100 | - | 64.1 | - |
| flip, crop, color | Mask2former [4] | Swin-B | 100 | - | 65.2 | +1.1 |
| crop, DatasetDM | Mask2former [4] | Swin-B | 100 | 40k (R:100) | 77.8 | +13.7 |
| flip, crop, color, DatasetDM | Mask2former [4] | Swin-B | 100 | 40k (R:100) | 78.5 | +14.4 |
| crop | DeepLabV3+ [3] | Mobilenet | 100 | - | 39.1 | - |
| crop, DatasetDM | DeepLabV3+ [3] | Mobilenet | 100 | 40k (R:100) | 45.3 | +6.2 |
| flip, crop, color | DeepLabV3+ [3] | Mobilenet | 100 | - | 40.5 | +1.4 |
| flip, crop, color, DatasetDM | DeepLabV3+ [3] | Mobilenet | 100 | 40k (R:100) | 46.1 | +7.0 |
| crop | DeepLabV3+ [3] | R50 | 100 | - | 45.1 | - |
| crop, DatasetDM | DeepLabV3+ [3] | R50 | 100 | 40k (R:100) | 55.3 | +10.2 |
| flip, crop, color | DeepLabV3+ [3] | R50 | 100 | - | 46.3 | +1.2 |
| flip, crop, color, DatasetDM | DeepLabV3+ [3] | R50 | 100 | 40k (R:100) | 56.9 | +11.8 |

Table 4: **Comparison with Data Augmentation.** 'R: ' refers to the number of real data used to train. 'crop', 'flip', and 'color' refer to the 'random crop', 'random horizontal flip', and 'color augmentation', respectively.

generate a total of 40k synthetic images for 20 categories. Subsequently, we utilize these data to train the Mask2former model and evaluate its performance on the 20 categories of VOC 2012.

- **Depth Estimation.** DepthFormer [13][3], serving as the baseline, is employed to assess our approach. We adhere to all network settings, loss functions, configurations, and training strategies outlined in the original implementation.

- **Pose Estimation.** We adopt HRNet [20] and its official code[4] for evaluating the pose estimation task on synthetic data generated by DatasetDM. Currently, we focus on single-person scenarios in each synthetic image and guide GPT-4 to generate corresponding images accordingly.

## 1.3 Training Setup for DatasetDM

All experiments for training DatasetDM were carried out on a single V100 GPU, while downstream task baselines (*i.e.* Mask2former, Depthformer) were trained using 8 V100 GPUs. Training our DatasetDM for $50k$ iterations with just one V100 GPU takes merely a day, showcasing its efficacy and efficiency. For all tasks, we employ the Adam optimizer [17] with a learning rate of 0.001 and a batch size of 1. The loss function and data augmentations vary for different tasks.

- **Semantic/Instance Segmentation.** During the training phase of DatasetDM, we primarily utilize two data augmentation techniques: random cropping to a size of 512×512 pixels, and random scaling.

---

[3] https://github.com/zhyever/Monocular-Depth-Estimation-Toolbox
[4] https://github.com/HRNet/HRNet-Human-Pose-Estimation

| method | backbone | # real image | # synthetic image | Sampled Classes for Comparison/% | | | | | mIoU |
|--------|----------|--------------|-------------------|-------|-------|----------|------|-------|------|
| | | | | outer | dress | headwear | belt | socks | |
| Baseline | R50 | 100 | - | 58.2 | 65.2 | 19.2 | 24.3 | 0 | 31.2 |
| **DatasetDM** (ours) | R50 | - | 40k (R:100) | 53.1 | 57.2 | 0.4 | 0.4 | 0 | 28.9 |
| **DatasetDM** (ours) | R50 | 100 | 40k (R:100) | 53.1 | 59.3 | 34.3 | 59.1 | 3.2 | 36.7 |
| Baseline | Swin-B | 100 | - | 58.1 | 56.1 | 64.3 | 33.4 | 7.2 | 40.1 |
| **DatasetDM** (ours) | Swin-B | 100 | 40k (R:100) | 70.0 | 70.8 | 72.0 | 32.8 | 5.9 | 45.1 |

Table 5: **Semantic segmentation on DeepFashion-MM [11].** 'R: ' refers to the number of real data used to train.

| method | backbone | # real image | # synthetic image | $\delta_1\uparrow$ | $\delta_2\uparrow$ | $\delta_3\uparrow$ | REL$\downarrow$ | Sq REL$\downarrow$ | RMS$\downarrow$ | RMS log$\downarrow$ |
|--------|----------|--------------|-------------------|-----------|-----------|-----------|------|---------|------|----------|
| Baseline | Swin-L | 50 | - | 0.59 | 0.84 | 0.93 | 0.31 | 0.37 | 0.81 | 0.30 |
| **DatasetDM** | Swin-L | - | 35k (R:50) | 0.68 | 0.90 | 0.97 | 0.22 | 0.19 | 0.60 | 0.23 |
| **DatasetDM** | Swin-L | 50 | 35k (R:50) | 0.68 | 0.91 | 0.98 | 0.21 | 0.18 | 0.63 | 0.23 |
| Baseline | Swin-L | 250 | - | 0.79 | 0.96 | 0.99 | 0.16 | 0.11 | 0.51 | 0.19 |
| **DatasetDM** | Swin-L | - | 35k (R:250) | 0.78 | 0.96 | 0.99 | 0.17 | 0.11 | 0.52 | 0.19 |
| **DatasetDM** | Swin-L | 250 | 35k (R:250) | 0.80 | 0.97 | 0.99 | 0.14 | 0.09 | 0.47 | 0.18 |

Table 6: **Depth Estimation on NYU Depth V2 `val` dataset.** Measurements are made for the depth range from $0m$ to $10m$.

- **Depth Estimation.** For depth estimation, we employ four data augmentation methods: random flipping, cropping, brightness-contrast adjustment, and hue-saturation value manipulation.
- **Pose Estimation.** For pose estimation, we use four data augmentation techniques: random scaling, cropping, flipping, and rotation.

## 1.4 Details for Training Data of DatasetDM

**Quantities of training read data.** Table 1 provides a comprehensive comparison of the quantities of training read data and synthetic data used for each downstream task in this study. Notably, with the exception of the seen class in the zero-shot segmentation setting, training with our DatasetDM requires less than $1\%$ of the available real data. This efficiency potentially reduces the implementation costs of perception algorithms and significantly improves data utilization.

# 2 Experiments

## 2.1 Comparison with Other Data Augmentation Methods.

From a certain perspective, the proposed DatasetDM is more akin to an efficient data augmentation technique, and thus we compare it with some previous data augmentation schemes, as shown in Table 4. Compared with flip and color augmentation, DatasetDM demonstrates a substantial advantage, bringing significant improvements, around $10\%$ increase, which is significant for the computer vision community.

## 2.2 Ablation Study for Baseline of Downstream Tasks.

In addition, the synthetic data generated by DatasetDM can seamlessly integrate with any existing downstream task model. To substantiate this claim, we tested our model with several other benchmark models, such as DeepLabV3, the results of which are detailed in Table 4. Notably, our synthetic data was able to enhance the performance of DeepLabV3 by approximately $10\%$, underscoring the robustness of our approach.

## 2.3 Human Pose Estimation on COCO `val2017`.

Table 2 presents the results of human pose estimation on the COCO 2017 dataset. Following the approach of HRNet [20], we established three distinct experimental settings, including variations in the backbone and input size, to evaluate the synthetic data from our model. Irrespective of the specific setting, our method consistently achieved an improvement of approximately $5\%$ in Average Precision

| methods | backbone | Train Set/% | | | mIoU/% | | |
| | | # real image | # synthetic image | categories | seen | unseen | harmonic |
|---|---|---|---|---|---|---|---|
| ZS3 [2] | - | 10.6k | - | 15 | 78.0 | 21.2 | 33.3 |
| CaGNet [10] | - | 10.6k | - | 15 | 78.6 | 30.3 | 43.7 |
| Joint [1] | - | 10.6k | - | 15 | 77.7 | 32.5 | 45.9 |
| STRICT [18] | - | 10.6k | - | 15 | 82.7 | 35.6 | 49.8 |
| SIGN [5] | - | 10.6k | - | 15 | 83.5 | 41.3 | 55.3 |
| ZegFormer [7] | - | 10.6k | - | 15 | 86.4 | 63.6 | 73.3 |
| Li *et al.* [14] | ResNet101 | | 10.0k (R:110k, COCO) | 15+5 | 62.8 | 50.0 | 55.7 |
| DiffuMask [21] | ResNet101 | - | 200.0 k (R:0) | 15+5 | 62.1 | 50.5 | 55.7 |
| DiffuMask [21] | Swin-B | - | 200.0k (R:0) | 15+5 | 71.4 | 65.0 | 68.1 |
| DatasetDM | ResNet101 | - | 40k (R:450, VOC) | 15+5 | 65.1 | 51.1 | 57.1 |
| DatasetDM | Swin-B | - | 40k (R:450, VOC) | 15+5 | 78.8 | 60.5 | 68.4 |

Table 7: **Zero-Shot Semantic Segmentation on PASCAL VOC 2012.** 'Seen', 'Unseen', and 'Harmonic' denote the mIoU of seen, and unseen categories, and their harmonic mean. 'R: ' refers to the number of real data from VOC 2012 or COCO 2017 used to train the generation model.

| methods | baseline | backbone | Train Set/% | | | mIoU |
| | | | # labeled real image | # unlabeled synthetic image | # synthetic image | |
|---|---|---|---|---|---|---|
| ReCo [16] | DeepLabv3 | R101 | 60 | 10.6k-60 | 0 | 53.3 |
| ReCo [16] | DeepLabv3 | R101 | 200 | 10.6k-200 | 0 | 69.8 |
| ReCo [16] | DeepLabv3 | R101 | 600 | 10.6k-600 | 0 | 72.8 |
| DatasetDM | DeepLabv3 | R101 | 60 | 0 | 40k | 57.6 |
| DatasetDM | Mask2former | R50 | 100 | 0 | 40k | 66.1 |
| DatasetDM | Mask2former | Swin-B | 100 | 0 | 40k | 78.5 |

Table 8: **Comparisons with semi-supervised works on PASCAL VOC 2012.**

(AP), which is a significant increase. Finally, it is noteworthy that our model attained competitive performance, with an AP of $48.9\%$, using merely 800 training images.

## 2.4 Semantic segmentation on DeepFashion-MM.

Table 5 showcases the performance of semantic segmentation on the DeepFashion-MM dataset [11]. Like our other experiments, we have conducted two sets of experiments using different backbones. Regardless of the setup, the joint training with synthetic data consistently outperforms the baseline that uses purely synthetic data, with an approximate improvement of $5\%$ mIoU.

## 2.5 Depth Estimation on NYU Depth V2 val dataset.

Table 6 presents the depth estimation experiment conducted on the NYU Depth V2 validation dataset [19]. Two training strategies have been devised based on variations in the training data. Independent of the data volume utilized, our approach consistently yields substantial improvements, specifically $0.1$ and $0.02$ respectively.

## 2.6 Zero-Shot Semantic Segmentation on VOC 2012

Consistent with preceding studies [14, 21], we conduct an experiment on zero-shot (open-vocabulary) semantic segmentation tasks using the VOC 2012 dataset [8]. Table 7 offers a comparative analysis with existing approaches to zero-shot semantic segmentation. In this experiment, our model is trained on a mere 450 images, with 30 images allocated for each of the 15 seen classes, and testing is conducted across all 20 categories. Despite the limited dataset in comparison to the complete set of 10.6k images, our model continues to exhibit competitive performance. In relation to methods employing synthetic data, our model achieves state-of-the-art (SOTA) performance, reaching $68.4\%$ mIoU.

## 2.7 Domain Generalization across Different Domains

Following DiffuMask [21], we further assess the domain generalization capabilities of synthetic data produced by DatasetDM, as depicted in Fig. 9. When compared with the previous state-of-the-art (SOTA) method, DatasetDM demonstrates superior effectiveness in domain generalization. For

| | | mIoU/% | | | |
|---|---|---|---|---|---|
| Train Set | Test Set | Car | Person | Motorbike | mIoU |
| Cityscapes [6] | VOC 2012 [8] `val` | 26.4 | 32.9 | 28.3 | 29.2 |
| ADE20K [22] | VOC 2012 [8] `val` | 73.2 | 66.6 | 64.1 | 68.0 |
| DiffuMask [21] | VOC 2012 [8] `val` | 74.2 | 71.0 | 63.2 | 69.5 |
| DatasetDM | VOC 2012 [8] `val` | 77.9 | 72.9 | 70.1 | 73.6 |
| VOC 2012 [8] | Cityscapes [6] `val` | 85.6 | 53.2 | 11.9 | 50.2 |
| ADE20K [22] | Cityscapes [6] `val` | 83.3 | 63.4 | 33.7 | 60.1 |
| DiffuMask [21] | Cityscapes [6] `val` | 84.0 | 70.7 | 23.6 | 59.4 |
| DatasetDM | Cityscapes [6] `val` | 85.6 | 58.9 | 12.7 | 52.4 |

Table 9: **Performance for Domain Generalization between different datasets.** Mask2former [4] with ResNet50 is used as the baseline. `Person` and `Rider` classes of Cityscapes [6] are consider as the same class, *i.e.*, `Person` in the experiment.

instance, DatasetDM achieves a score of 73.6%, as opposed to DiffuMask's score of 69.5% on the VOC 2012 `val` set. Compared to real data, DatasetDM exhibits enhanced robustness in terms of generalization. It is reasonable that synthetic data exhibits greater diversity, especially when integrated with language models, as shown in Fig. 4 In terms of diversity and robustness, it far surpasses real datasets.

## 2.8 Comparison with the semi-supervised approaches on VOC2012

Table 8 presents the comparison between the prior semi-supervised works and our DatasetDM. Even with a smaller amount of data (60 images), our approach demonstrates competitive performance, outperforming current semi-supervised semantic segmentation works. Furthermore, with a more powerful backbone, our method can achieve even better performance, reaching a mIoU of 78.2 with only 100 images.

## 2.9 More Qualitative Results

To demonstrate the high-quality synthetic data, we visualized synthetic data from two domains: human-centric and urban city, as shown in Fig. 1 (human-centric) and Fig. 2 (urban city scenario). The human-centric domain predominantly encompasses datasets related to human activity, such as COCO 2017, Cityscapes, and DeepFashion-MM. On the other hand, the urban city scenario pertains specifically to datasets like Cityscapes and COCO 2017. To the best of our knowledge, our work is the first to support multi-task synthesis of data. We believe that unified annotation synthesis is meaningful and can support interactions between different modalities. Recent works, *e.g.* ImageBind [9] have already demonstrated its feasibility and necessity. Our method also has many advantages, such as the ability to custom design datasets for a specific domain or to address bad case scenarios, and it is particularly effective in solving problems related to long-tail data distribution. This is straightforward; we can achieve it simply by adjusting our prompts.

# 3 Details on the Architecture of Perception Decoder

We show the detailed architecture of our P-Decoder in Fig. 3, which consists of pixel decoder, text encoder, transformer decoder.

## 3.1 Text Encoder for Open-Vocabulary Segmentation.

In the open-vocabulary setting, for each class, we encode the corresponding class name (*i.e. cat*, *dog*) into a d-dimensional vector using the CLIP encoder. For a word corresponding to two text tokens, we average them into one token. Subsequently, this token is replicated $n$ times, resulting in an $n \times d$ matrix. The matrix is then concatenated with a learnable query embedding of dimensions $n \times 768$. Ultimately, the concatenation is processed through a Multilayer Perceptron (MLP) layer to fuse the elements.

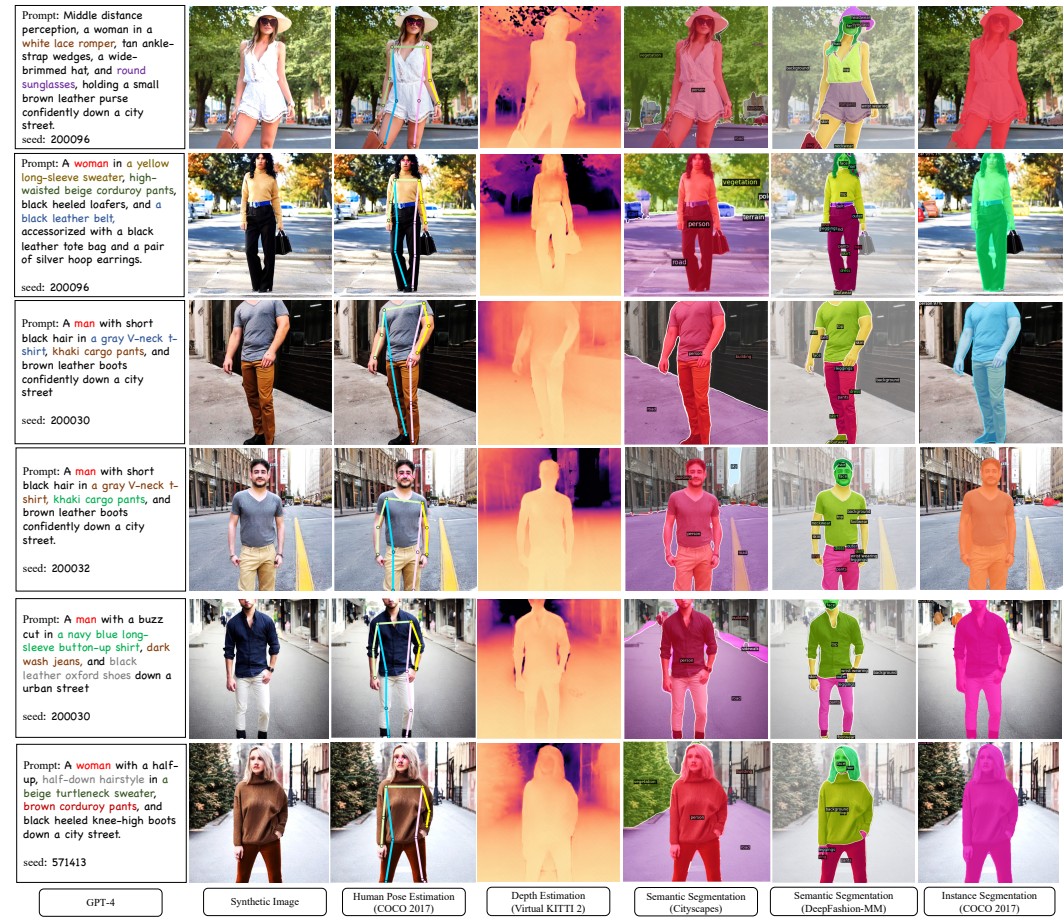

Figure 1: **Examples of Human-Centric Generated Data for DatasetDM.** Our method is capable of generating high-accuracy, high-diversity, and unified perceptual annotations.

## 3.2  Semantic and Instance Segmentation.

With the representation $\hat{\mathcal{F}}$, which is fused from multi-scale features and cross-attention maps, we employ a pixel decoder and a transformer decoder to derive the per-pixel embedding $C \times H \times W$ and mask embedding $C \times N$. As per the method outlined by Li.*et al* [14], the pixel decoder consists of several straightforward up-sampling layers. Each layer comprises four types of computations: 1) $1 \times 1$ `Conv` for adjusting feature dimensionality, 2) `Upsample` using simple linear interpolation to upscale the feature to a higher spatial resolution, 3) Concat for merging features from different layers, and 4) `Mix-conv` for blending features from varying spatial resolutions, which includes two $3 \times 3$ Conv. Similar to Mask2former [4], the transformer decoder comprises a stack of transformer layers with cross-attention, self-attention, and masked attention. The final mask predictions of dimensions $N \times H \times W$ can be obtained by performing a simple matrix multiplication of the per-pixel embedding of dimensions $C \times H \times W$ and the mask embedding of dimensions $C \times N$.

## 3.3  Human Pose and Depth Estimation.

By expanding the segmentation architecture with the addition of two convolutional layers to the pixel decoder, we are able to efficiently handle the associated tasks of pose and depth estimation. Consequently, we derive two predictive outputs, denoted by $\mathbf{O} \in {}^{M \times H \times W}$ and $\hat{\mathbf{O}} \in {}^{M \times H \times W}$, corresponding to the human pose and depth estimation tasks, respectively.

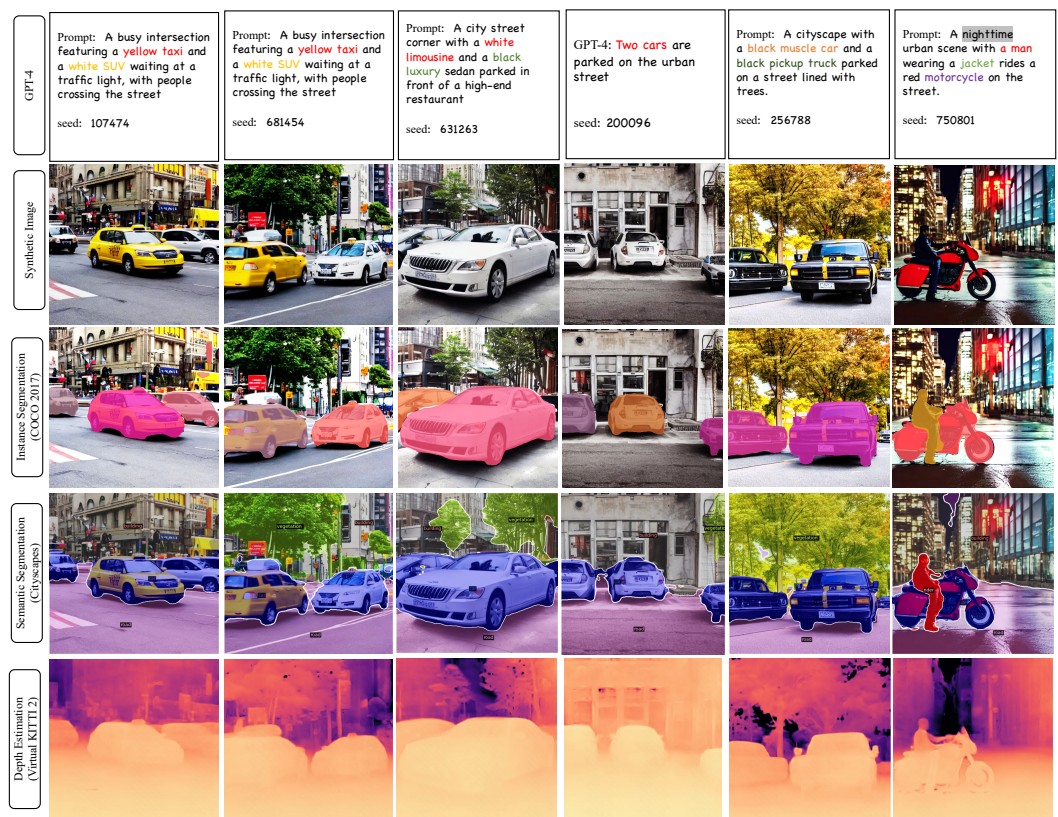

Figure 2: **Examples of Generated Data for Urban City Scenario from DatasetDM.**

# 4 Synthetic Dataset

## 4.1 Prompts from GPT-4

Here, we also demonstrate the detailed process of prompt generation, guided by GPT-4, as shown in Fig. 4. Throughout the process, human only need to provide a small number of prompts to guide GPT-4. With a cost of no more than 50 words prompt clue, we can accomplish the generation of a massive number of prompts for a downstream task dataset. It is worth mention that text-guided data is extremely flexible. We can customize the generation of certain attributes of data domain. For instance, if we need to enhance the variation in the number of objects, we can provide a prompt like `More variation in number`. This is extremely flexible and convenient.

## 4.2 Prompts for Each Dataset

As shown in Table 10, we also provide some prompt cases of our method for each dataset, and we will open-source these prompts along with the corresponding code. For tasks that distinguish between classes, *i.e.* semantic and instance segmentation, we will guide GPT-4 to generate around 100 descriptions specifically for each class. For tasks and datasets that are not class-sensitive, *e.g.* pose and depth estimation, we guide GPT-4 to generate a large number of descriptions all at once.

# 5 Limitation and Future Work

## 5.1 Potential Negative Societal Impacts

As with other projects involving synthetic image generation, the potential adverse societal implications of our work largely revolve around ethical considerations. Utilizing the Stable Diffusion model, trained on the 5-billion image LION dataset, raises notable private copyright concerns due to the nature

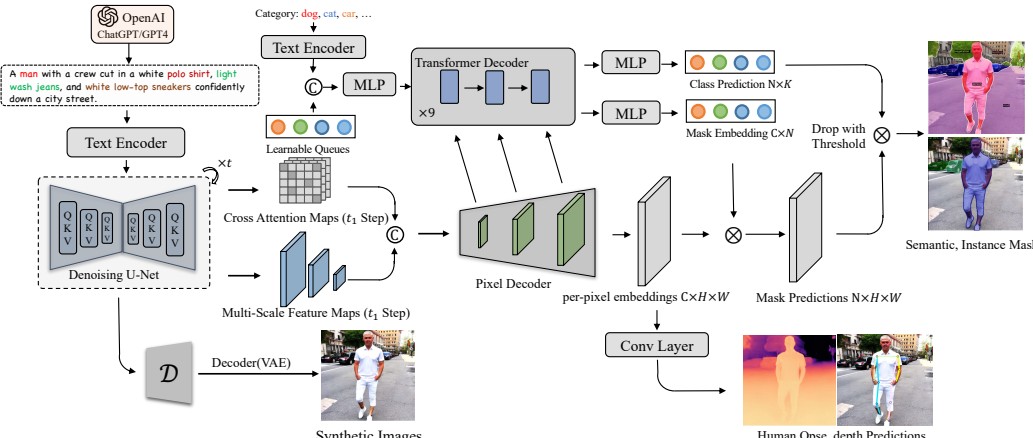

Figure 3: **Details for P-Decoder.** The whole framework of decoder consists of text encoder, pixel decoder, and transformer decoder. For different downstream task, we only need to adjust *minor variations i.e.* whether to startup some layers.

of the dataset. Nonetheless, we maintain that such potential repercussions should not overshadow the applicability and value of our research. Numerous text-guided text-to-image diffusion works exist, of which ours is but one. We also emphasize that it's feasible to mitigate these ethical issues through the careful crafting of specific prompts, serving as an effective countermeasure. Additionally, we can mitigate this impact from an algorithmic perspective [12], such as eliminating certain concepts (which may infringe on personal privacy) from the pretrained model.

## 5.2 Limitation & Future Work

**Limitation.** The main limitation of this study is that the quality and complexity of the synthesized data still cannot compare with real data. If certain companies and organizations could invest substantial resources to collect and manually annotate massive amounts of training data, better results could be achieved. However, this is actually the main limitation of the Stable diffusion model. We could also consider using more powerful diffusion models to alleviate this issue, as shown in Fig. 5. Deepfloyd IF [5] is a more powerful text-guided image generation model, which significantly outperforms Stable diffusion in two main aspects. First, it excels in semantic alignment - given a lengthy text description, the IF model can generate related images more accurately. Second, the IF model can synthesize images at a higher resolution, specifically 1024, while the resolution of Stable diffusion is only 512. We believe that our method, in combination with the DeepFloyd IF model, can lead to further improvements and make a greater contribution. Due to time constraints (DeepFloyd IF released on May 2023), we are unable to provide related experiments, but this does not affect the validation of the effectiveness of our method. Our primary contribution lies in using a unified decoder to parse the latent space of the pre-trained diffusion model, not in enhancing the quality of image synthesis.

**Future Work** This study is intriguing and innovative, possessing profound exploratory significance. We identify several avenues for future enhancement: firstly, employing a more robust text-guided image generation model may yield substantial improvements. Secondly, augmenting the efficiency of prompt generation, or designing prompts that better align with the target domain could prove beneficial. For example, synthesizing specific prompts corresponding to the COCO 2017 dataset could be viable.

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

| Dataset | Category | Prompts |
|---------|----------|---------|
| VOC 2012 [8] | Car | A classic red convertible parked near a sandy beach, its vibrant color contrasting with the clear blue sky.
... |
| | Person | A young woman jogging in a park, wearing athletic clothing and listening to music through her earphones.
... |
| | Dog | A playful Golden Retriever, its fur gleaming in the sunlight, splashing in the water at a dog-friendly beach.
... |
| Cityscapes [6] | Car | A sleek, black car cruises down a busy urban street lined with towering skyscrapers.
... |
| | Person | A woman in a red dress is crossing the street at the crosswalk while cars wait for her.
... |
| | Bus | A red double-decker bus drives through the heart of the city on a busy urban street, the passengers admiring the sights from the upper level.
... |
| COCO 2017 [15] | Car | Two red cars parked on a busy city street in the afternoon.
... |
| | Person | A group of people playing volleyball on a beach, with the ocean in the background.
... |
| | Bus | A big red bus parked on the side of the road with a tree behind it.
... |
| NYU Depth V2 [19] | - | A kitchen with white cabinets, stainless steel appliances, and a wooden table.
A bedroom with a queen-size bed, dresser, and nightstand.
... |
| COCO 2017 Pose [15] | - | a person with a backpack, wearing a green jacket and khaki pants.
a middle-aged woman wearing a red blazer, black slacks, and pumps.
... |
| DeepFashion-MM [11] | - | A woman wearing a loose-fitting white blouse with ruffled sleeves, paired with high-waisted, wide-leg navy blue pants and black ankle-strap stiletto heels.
A man dressed in a classic white button-up shirt, khaki chinos with a slim fit, and brown suede desert boots.
... |

Table 10: **Prompts for Different Datasets.** For different data domains, we will guide GPT-4 to generate corresponding prompts. We will release the code and corresponding prompts files.

[3] L.-C. Chen, Y. Zhu, G. Papandreou, F. Schroff, and H. Adam. Encoder-decoder with atrous separable convolution for semantic image segmentation. In *Proceedings of the European conference on computer vision (ECCV)*, pages 801–818, 2018.

[4] B. Cheng, I. Misra, A. G. Schwing, A. Kirillov, and R. Girdhar. Masked-attention mask transformer for universal image segmentation. In *Proceedings of the IEEE/CVF Conference on Computer Vision and Pattern Recognition*, pages 1290–1299, 2022.

[5] J. Cheng, S. Nandi, P. Natarajan, and W. Abd-Almageed. Sign: Spatial-information incorporated generative network for generalized zero-shot semantic segmentation. In *Proc. ICCV*, 2021.

[6] M. Cordts, M. Omran, S. Ramos, T. Rehfeld, M. Enzweiler, R. Benenson, U. Franke, S. Roth, and B. Schiele. The cityscapes dataset for semantic urban scene understanding. In *Proceedings of the IEEE conference on computer vision and pattern recognition*, pages 3213–3223, 2016.

[7] J. Ding, N. Xue, G.-S. Xia, and D. Dai. Decoupling zero-shot semantic segmentation. In *Proc. CVPR*, 2022.

[8] M. Everingham, L. Van Gool, C. K. Williams, J. Winn, and A. Zisserman. The pascal visual object classes (voc) challenge. *International journal of computer vision*, 88(2):303–338, 2010.

[9] R. Girdhar, A. El-Nouby, Z. Liu, M. Singh, K. V. Alwala, A. Joulin, and I. Misra. Imagebind: One embedding space to bind them all. In *CVPR*, 2023.

[10] Z. Gu, S. Zhou, L. Niu, Z. Zhao, and L. Zhang. Context-aware feature generation for zero-shot semantic segmentation. In *ACM MM*, 2020.

[11] Y. Jiang, S. Yang, H. Qiu, W. Wu, C. C. Loy, and Z. Liu. Text2human: Text-driven controllable human image generation. *ACM Transactions on Graphics (TOG)*, 41(4):1–11, 2022.

[12] N. Kumari, B. Zhang, S.-Y. Wang, E. Shechtman, R. Zhang, and J.-Y. Zhu. Ablating concepts in text-to-image diffusion models. *arXiv preprint arXiv:2303.13516*, 2023.

[13] Z. Li, Z. Chen, X. Liu, and J. Jiang. Depthformer: Exploiting long-range correlation and local information for accurate monocular depth estimation. *arXiv preprint arXiv:2203.14211*, 2022.

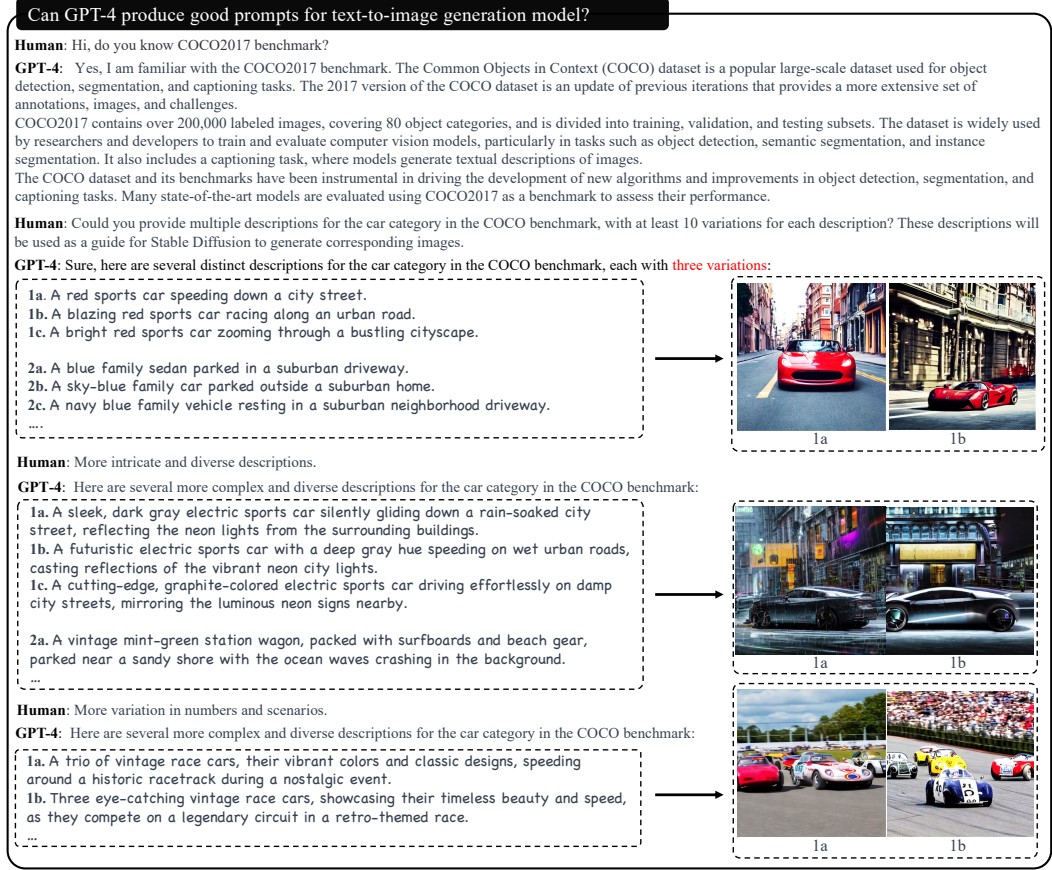

Figure 4: **Prompts of diffusion model from GPT-4.** By providing some simple cues, GPT-4 can generate a vast and diverse array of prompts.

[14] Z. Li, Q. Zhou, X. Zhang, Y. Zhang, Y. Wang, and W. Xie. Guiding text-to-image diffusion model towards grounded generation. *arXiv preprint arXiv:2301.05221*, 2023.

[15] T.-Y. Lin, M. Maire, S. Belongie, J. Hays, P. Perona, D. Ramanan, P. Dollár, and C. L. Zitnick. Microsoft coco: Common objects in context. In *Computer Vision–ECCV 2014: 13th European Conference, Zurich, Switzerland, September 6-12, 2014, Proceedings, Part V 13*, pages 740–755. Springer, 2014.

[16] S. Liu, S. Zhi, E. Johns, and A. J. Davison. Bootstrapping semantic segmentation with regional contrast. *arXiv preprint arXiv:2104.04465*, 2021.

[17] I. Loshchilov and F. Hutter. Decoupled weight decay regularization. *arXiv preprint arXiv:1711.05101*, 2017.

[18] G. Pastore, F. Cermelli, Y. Xian, M. Mancini, Z. Akata, and B. Caputo. A closer look at self-training for zero-label semantic segmentation. In *Proc. CVPRW*, 2021.

[19] N. Silberman, D. Hoiem, P. Kohli, and R. Fergus. Indoor segmentation and support inference from rgbd images. *ECCV (5)*, 7576:746–760, 2012.

[20] K. Sun, B. Xiao, D. Liu, and J. Wang. Deep high-resolution representation learning for human pose estimation. In *Proceedings of the IEEE/CVF conference on computer vision and pattern recognition*, pages 5693–5703, 2019.

[21] W. Wu, Y. Zhao, M. Z. Shou, H. Zhou, and C. Shen. Diffumask: Synthesizing images with pixel-level annotations for semantic segmentation using diffusion models. *arXiv preprint arXiv:2303.11681*, 2023.

[22] B. Zhou, H. Zhao, X. Puig, S. Fidler, A. Barriuso, and A. Torralba. Scene parsing through ade20k dataset. In *Proceedings of the IEEE conference on computer vision and pattern recognition*, pages 633–641, 2017.

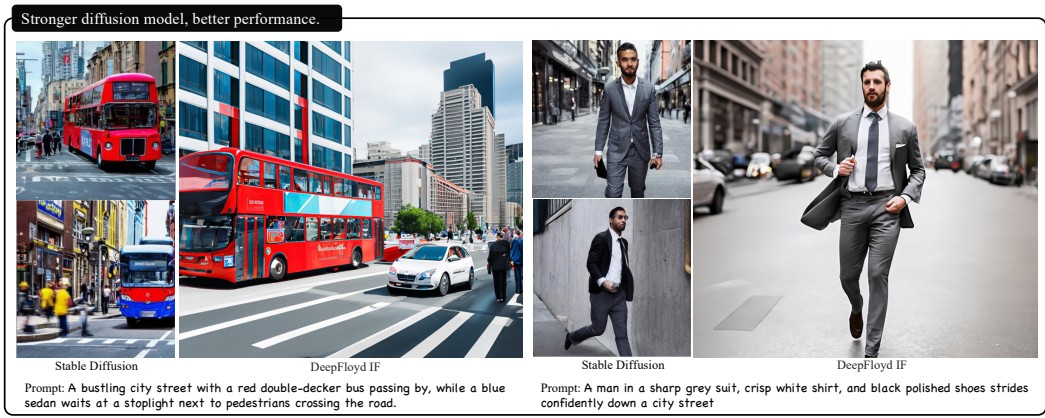

Figure 5: **Stronger Diffusion Model, Greater Potential.** With the advancements in generative models, synthetic data will have greater potential and possibilities for perception tasks. A simple solution is to replace Stable Diffusion with DeepFolyd IF directly.