# OpenReview forum: "DatasetDM: Synthesizing Data with Perception Annotations Using Diffusion Models"
_NeurIPS.cc/2023/Conference — NeurIPS 2023 poster_

### Official Review · Reviewer_aPc6 · 2023-07-05

**Soundness:** 2 fair
**Presentation:** 3 good
**Contribution:** 2 fair
**Rating:** 5
**Confidence:** 4

**Summary:**

The paper proposes a new framework called DatasetDM that leverages the rich visual knowledge contained in diffusion models to generate synthetic data and corresponding annotations for various perception tasks, The core component of DatasetDM is the P-Decoder for generating perception annotations. The primary process is as follows: 1) During the training stage, a pre-defined text template is employed to extract feature maps and cross-attention maps from the pre-trained Stable Diffusion UNet, which are then concatenated to create a feature representation. The feature representation is then utilized to train P-Decoder, which employs a unified architecture that can adapt to various perception tasks; 2)During the generation stage, random sampling is conducted from a vast quantity of prompts provided by GPT-4 to enhance diversity. Subsequently, new images are generated by the VAE decoder of vanilla Stable Diffusion, while corresponding annotations are generated by the P-Decoder. The synthetic data and annotations will be utilized for various downstream tasks.

**Strengths:**

The paper proposes a novel approach to leveraging the visual representations contained in the pre-trained Stable Diffusion for solving multiple perception tasks. The methodology is well-organized and clearly illustrated. Notably, there is limited research focusing on the utilization of the rich latent space in diffusion models for downstream vision tasks. Thus, the methodology presented in the paper is informative and enlightening.

**Weaknesses:**

1) The methodology lacks some novelty, as it bears a resemblance to the paper VPD[a]. Specifically, the method employed in DatasetDM for extracting visual representations from Stable Diffusion is nearly identical to that of VPD and only with minor differences in implementation details, including using a similar technique to extract multi-scale feature maps and cross-attention maps and employing the same manipulation of concatenation. Additionally, DatasetDM adopts a resembling design to VPD, as VPD also employs task-specific decoders for perception tasks and also uses a unified architecture with minor task-specific differences for the architecture design of the decoder.
2) The experiments are insufficient. Firstly, to further illustrate the effectiveness of synthetic data, an ablation on the number of synthetic images can be conducted, especially investigating the performance of using a small number of synthetic images(e.g., 400 real images and 400 synthetic images versus 400 real images only). Secondly, more SOTA methods can be included as baseline to further evaluate the effectiveness of DatasetDM. Furthermore, evaluation on some less common datasets is necessary, which are more likely to face issues of data and annotation scarcity
[a] "Unleashing Text-to-Image Diffusion Models for Visual Perception"(VPD)


**Questions:**

1) The paper requires further clarification on how the correspondence between generated images and annotations is ensured. Specifically, within the framework of DatasetDM, new images are generated by the VAE decoder of vanilla Stable Diffusion, while perception annotations are generated by the proposed P-Decoder， which utilizes a distinct architecture and does not share the same input with the former.

2) Please further clarify why training the P-Decoder only requires less than 1% of manually labeled images. Visual align/instruct tuning is listed in the contribution, while not elaborated in the methodology.

3) It is advised to utilize appropriate evaluation metrics or conduct further visualization experiments to provide an intuitive observation of the quality of synthetic data and the accuracy of the annotations.

4) Sometimes Stable Diffusion may encounter difficulty in comprehending the given text prompt, resulting in unexpected image generation, I am curious whether such instances occurred during the experiments. Furthermore, is there a potential for repetition in the vast quantity of prompts generated by GPT-4? As In the ablation study, the performance improvement appears to plateau when the number of prompts is sufficiently large.

5) There are spelling errors in Fig 2 a) and Fig 4, where "DatasetDB" should be corrected to "DatasetDM".


**Limitations:**

The limitations of the paper are not explicitly addressed, and it would be beneficial to add a section about limitations.

---

> ### Author Rebuttal · Authors · 2023-08-06
>
> Thanks to the reviewers for the valuable comments.
>
> **Weakness 1**: Comparison with VPD.
> **R1**: We want to emphasize that our work and VPD (unpublished) are concurrent works. And we argue that our work is not a resemblance for the following reasons:
>
> - **Extracting features.**  Using the diffusion model to extract features is a common operation, and VPD is not the first work to do so. Previous works[1][2] have also employed this approach, similar to using a backbone to extract features. The contribution of this paper (L69-85) lies in the design of the versatile dataset generation model.
>
> - **Cross-attention.** We did not claim that the cross attention is our contribution (L69-85), while its performance has already been validated by VPD and DiffuMask[3]. We have cited VPD and mentioned it in the context of the cross-attention maps (L178-180).
>
> - **Perception decoder**. The two decoders are completely different. We design a transformer-based architecture (Fig. 3), while VPD uses existing decoder models (CNN-based), i.e., FPN. VPD does not support instance segmentation, as it predicts a single mask for multi-class classification. Meanwhile, we support a broader range of tasks, including pose estimation, zero-shot, and long-tail segmentation, which can not be supported by VPD. After carefully reviewing the VPD paper and official coding, we are confident in this observation.
>
> - **Training**. We explore few-shot tuning by generating more synthetic data for auxiliary training. VPD, on the other hand, focuses on training with the entire dataset.
>
> - **Different Task**. Our work focuses on synthetic data augmentation, while VPD is a perception model.
>
> **Summary**: Overall, we have `numerous differences` with VPD for the core `contribution` (L69-85 in our paper), with the most significant being the dissimilarity in decoder structure and the supported tasks. VPD can not support for instance segmentation, pose estimation, zero-shot segmentation, and long-tail segmentation.
>
> **Weakness 2-1**: Ablation on the number of synthetic images. **R2**: See Tab. 4 of  the rebuttal PDF for the ablation study on VOC 2012. Even using only 100 synthetic images (equivalent to the number of real images), we can still achieve a `2.6%` mIoU improvement. With 400 images, a significant improvement of `10.9%` can be achieved.
>
> Additional points worth mentioning include: **Computing cost.** The only difference between 100, 400, and 40k synthetic images is the inference computational cost, which ranges from a few minutes to half a day. In practical applications, there is no reason to limit the use of only 100 images. **Compare with the prior[2][3][4].** Previous highly relevant works, such as DiffuMask[3], and HandsOff[4], all generate at least thousands of synthetic data.
>
> **Weakness 2-2**: More SOTA methods. **R3**: We have provided extensive experiments with **three** SOTA methods (L244-L259)  across **six** different tasks. Here we additionally provide a pose estimation method, SimCC[5], on the COCO val2017 dataset(see Tab. 7 of the rebuttal PDF)
>
>
> **Weakness 2-3**: Less common datasets. **R4**: We are conducting relevant experiments on LVIS[10] and will strive to provide before the rebuttal deadline .
>
> **Q 1**: Correspondence between images and annotations. **R5**:  The inputs of VAE and decoder belong to the same set of features. The decoder utilizes $F$ (L172), which are the Multi-Scale Feature Maps of the UNet (see Fig. 2). The feature $F$ with $64\times64$ resolution at the 1000-th denoising step also is the input for the VAE decoder.
>
> **Q 2-1**: Only requires less than 1% of data. **R6**:  Because the used Stable Diffusion trained on 5 billion data (LAION5B, L48), it possesses strong semantic understanding capabilities, similar to the principles behind using language models(GPT-3[8], Self-instruct[9]) for aligning/instruct tuning, where large models exhibit robust generalization abilities.
>
> **Q 2-2**: Visual align/instruct tuning. **R7**: The tuning is more like a training strategy where pre-trained large models[8][9] on massive datasets exhibit strong generalization abilities, and for a specific task, only a small amount of task-specific data is needed to achieve good results. In our work (Fig. 2), we only need to train the decoder.
>
> **Q 3**:  Visualization. **R8**: Bad cases see Fig. 1 and Fig. 2 of rebuttal PDF, good cases see Fig. 4, Fig. 5 of paper, and Fig. 1, Fig. 2 of the supplementary material.
>
> **Q 4-1**:  Unexpected generation from prompt. **R9**: Good question. The phenomenon exists. We have discussed this in the Supplementary Material, Sec. 5.2, and Fig. 5, where providing a prompt `a blue sedan waits at a stoplight` resulted in Stable Diffusion not generating the blue sedan. One solution is to use more powerful generation models, such as IF[7].
>
> **Q 4-2**:  Repetitive prompt. **R10**: Almost no repeated prompts since we only generate 100 prompts for each category (L227). The performance plateau is determined by various factors, including the quality of images and the accuracy of masks. Therefore, increasing prompts may not necessarily improve the performance plateau.
>
> [1] Open-vocabulary panoptic segmentation with text-to-image diffusion models.CVPR. 2023.
>
> [2] Guiding text-to-image diffusion model towards grounded generation." arXiv preprint (2023).
>
> [3] Diffumask: Synthesizing images with pixel-level annotations for semantic segmentation using diffusion models. arXiv preprint (2023).
>
> [4] HandsOff: Labeled dataset generation with no additional human annotations. CVPR. 2023.
>
> [5] Simcc: A simple coordinate classification perspective for human pose estimation." ECCV, 2022.
>
> [6] lvis: a dataset for large vocabulary instance segmentation.
>
> [7] deep-floyd/IF
>
> [8] Language models are few-shot learners." (2020): 1877-1901.
>
> [9] Self-instruct: Aligning language model with self generated instructions." arXiv preprint (2022).
>
> [10] lvis: a dataset for large vocabulary instance segmentation.

---

> > ### Author Response · Authors · 2023-08-16
> > **Further feedback**
> >
> > **Weakness 2-3**: Less common datasets.
> >
> > **R4**: We provide results on a less common dataset, LVIS[1], which encompasses 1203 categories and is commonly employed to assess long-tail distribution of instance segmentation tasks. We trained on 1203 real images (one image per class), generating an additional 50 synthetic images per class for augmentation. This approach yielded approximately a 3% increase in AP.
> >
> >
> >  | labeled real image | synthetic images | AP|
> > |  ----  | ----  |----  |
> > | 1203 images(one image for one class) | 0 | 8.45
> > | 1203 images(one image for one class) | 51k | 11.78
> >
> > Thank you sincerely for your valuable feedback on our submission. We have read your comments carefully and have addressed them in our rebuttal. As the second phase of the rebuttal process is approaching its conclusion, we kindly ask for your acknowledgment if our responses have effectively addressed your comments. We would also be happy to engage in further discussions if needed. Thank you again for your time and consideration.
> >
> > [1] lvis: a dataset for large vocabulary instance segmentation.

---

> ### Comment · Reviewer_aPc6 · 2023-08-17
>
> Thanks the authors for the clarification, I will raise my score to 5. I suggest the authors add more discussion about VPD in the revision version.

---

> > ### Author Response · Authors · 2023-08-17
> > **Response to Reviewer aPc6**
> >
> > Dear Reviewer aPc6,
> >
> > Thank you for your suggestions and support regarding our work. VPD is indeed an excellent piece of research. We will incorporate more comparisons and descriptions related to VPD in the revised draft, particularly in Section 2 (Related Work) and other relevant sections.
> >
> > Best regards,
> >
> > Authors of paper 5020.

---

### Official Review · Reviewer_DG4j · 2023-07-06

**Soundness:** 3 good
**Presentation:** 3 good
**Contribution:** 1 poor
**Rating:** 4
**Confidence:** 5

**Summary:**

The paper introduces a novel dataset generation method producing diverse synthetic images along with perception annotations including semantic/instance segmentation, and depth/pose estimation. The technique leverages a pretrained text-to-image generation model, i.e., Stable Diffusion, to extract cross-attention maps and multi-scale feature maps as features input to multiple trainable decoders for different tasks. By training these add-on decoders on only a small number of labeled images (typically 100-800 images), the model can generate annotations along with the synthetic images. This method is evaluated on many tasks and achieves promising results.

**Strengths:**

+ This paper addresses an interesting and impactful problem.
+ The proposed method can generate pseudo-labels for several tasks segmentation, depth estimation, etc. with a small number of labeled images.


**Weaknesses:**

+ Limited novelty: the presented method is fairly similar to [a], published more than 2 months prior to this submission. The most significant difference is that in [a], the authors work directly on real images while this method synthesizes new images to train a corresponding network to produce prediction. No mentions or discussions are provided.
+ The current setting is equivalent to the semi-supervised segmentation, the authors should compare with several semi-supervised methods in the corresponding tested perception tasks.
+ The authors should include experiments to examine the contribution of cross–attention maps or multi-scale feature maps alone and in combination with ablation studies to analyze the contribution of each module to the overall performance.

Minor:
+ In L194, the authors use the notation Q for the learnable queues, however, in L207, the authors notated Q as query embedding. I would suggest using different Q_{x} for separate notations.
+ The mask generation process should be (briefly) discussed in the main paper instead of the supplementary documents.

[a] Zhao, Wenliang, et al. "Unleashing text-to-image diffusion models for visual perception." arXiv preprint arXiv:2303.02153 (2023).

---
After reading the rebuttal, the highest score that I can give for the paper is 4.

**Questions:**

+ After averaging the cross-attention maps within certain resolutions, do the authors fuse the averaged cross-attention maps with the corresponding feature maps in the same resolution or fuse two modalities in the same resolution first?
+ Refer to Sec. 3.2 in the supplementary document, how can the authors generate the semantic/instance masks with the class prediction and mask predictions? How can the authors choose the threshold to obtain the final masks?
+ Do the authors observe any failure cases when generating the annotations?
+ Does the proposed method work for all classes in PASCAL-VOC 2012 or COCO2017 (the authors can show the class-specific results)?


**Limitations:**

No discussion on the method's limitations was provided.

---

> ### Author Rebuttal · Authors · 2023-08-06
>
> Thanks to the reviewers for the valuable comments.
>
> **Weaknesses 1**: Comparison with VPD (novelty). **R 1**: We want to emphasize that our work and VPD[1] (unpublished) are concurrent efforts. Besides, we believe that our work and VPD are entirely different for the following reasons:
>
> - **Different Task**. Our work is a synthetic data augmentation for perception tasks, while VPD is a diffusion-based perception algorithm.  In addition, we support more tasks than VPD, including Pose Estimation, Fashion Segmentation, and Open-Vocabulary Segmentation, which are not merely simple extensions but require corresponding improvements in the perception decoder structure.
>
> - **Different Architecture**.  The most crucial design component in VPD or our work is the perception decoder, and in this aspect, we are completely different. We design a transformer-based approach to model multiple tasks (Figure 3), while VPD employs existing decoder models (CNN-based), such as FPN[6] for semantic segmentation, using another CNN-based decoder for referring segmentation. The task-specific decoder of VPD utilizes multiple architectures but does not support instance segmentation.
>
> - **Training strategy**. We explore few-shot tuning by generating more synthetic data for auxiliary training. VPD, on the other hand, focuses on training with the entire dataset. Our emphasis is on performance improvement under limited data cost, while VPD serves as a performance method with abundant data.
>
> - **No mentions or discussions**. We have cited VPD and mentioned it in the context of the Text-Image Representation Fusion (L178-180), which is the closest connection we have with VPD. We did not claim that the cross attention is our contribution (L69-85), while its performance has already been validated by VPD[1], DiffuMask[2], and P2P[3].
>
> **Summary**: Except for the feature extraction(common operations, similar to Prior[1][4][5]), and the usage of cross attention (not our contribution), all other parts (`task` and `decoder structure`) are completely different from VPD. Our `unique` feature (L69-85) is a versatile dataset generation model, where a transformer decoder can simultaneously generate various perception annotations.
>
> **Weaknesses 2**: Comparison with the semi-supervised segmentation. **R 2**: In some downstream tasks, we have provided some weakly supervised comparisons, including Zero-Shot Semantic Segmentation in Table 5 and Table 9 (supplementary material), where ZegFormer, and SIGN are weakly supervised methods.
>
> In addition, we would like to emphasize that the current setting is `different` from weakly supervised segmentation. Existing weakly supervised methods, such as Zegformer and SIGN, utilize the entire training set with 10k+ images for PASCAL-VOC 2012 and do not use mask annotations, whereas we only use 100 images and utilize mask annotations. Therefore, a direct comparison between the two settings is not appropriate.
>
> In comparison, some works for synthetic data augmentation (e.g., DiffuMask [2], Li et al. [5], and HandsOff [7]) are more reasonable for comparison. These methods all explore synthetic data augmentation for perception tasks and have similar settings. Especially HandsOff and our approach both explore training with limited data to generate more augmented data. Under the same setting, for Semantic segmentation on Cityscapes, we outperform HandsOff by **17.1%** mIoU (see Supplementary Material, Table 3).
>
> **Weaknesses 3**: Related experiments for cross–attention maps or multi-scale feature maps. **R 3**: We have provided the relevant experiments in Tab 4 (a) and (b), where the impact of using cross-attention maps or multi-scale feature maps with different steps on the results is shown. The results show that using the cross-attentions from appropriate step can lead to a gain of approximately 1%.
>
> **Question 1**: Details for fusing two modalities
> **R4**: We fuse features and cross-attention maps at different resolutions and then use them. Specifically, given four feature maps at different resolutions $\mathcal{F}_1, \mathcal{F}_2, \mathcal{F}_3, \mathcal{F}_4$, and four cross-attention maps at different resolutions $\mathcal{\hat{A}}_1, \mathcal{\hat{A}}_2, \mathcal{\hat{A}}_3, \mathcal{\hat{A}}_4$, we first fuse the features for each resolution. For the $8 \times 8$ resolution, we obtain $\mathcal{\hat{F}}_1=\text{Conv}
> ([\mathcal{F}_1,\mathcal{\hat{A}}_1])$. Then, we use the four final features ($\mathcal{\hat{F}}_1,\mathcal{\hat{F}}_2,\mathcal{\hat{F}}_3,\mathcal{\hat{F}}_4$) for the perception tasks.
>
> **Question 2**: How can the authors choose the threshold to obtain the final masks? **R5**: The process of mask generation is the same as in the baseline Mask2Former (L246), where we did not select specific mask thresholds. The thresholds for both instance/semantic masks and classification are set to 0.5.
>
> **Question 3**: Failure Cases. **R6**: We present some failure cases in the author rebuttal PDF (Fig. 1, Fig. 2).
>
> **Question 4**: Class-specific results for PASCAL-VOC 2012. **R7**: Please refer to the author rebuttal PDF for all 21 classes (Tab. 6).
>
> **Limitations 1**: No discussion of limitations. **R8**: We have already provided one section to discuss the limitations in Supplementary Material Sec. 5.2 (L207-221).
>
>
> [1] "Unleashing text-to-image diffusion models for visual perception." arXiv preprint  (2023).
>
> [2] "Diffumask: Synthesizing images with pixel-level annotations for semantic segmentation using diffusion models." arXiv preprint  (2023).
>
> [3] "Prompt-to-prompt image editing with cross attention control." arXiv preprint (2022).
>
> [4] "Open-vocabulary panoptic segmentation with text-to-image diffusion models." CVPR. 2023.
>
> [5] "Guiding text-to-image diffusion model towards grounded generation." arXiv preprint (2023).
>
> [6] "Panoptic feature pyramid networks." CVPR. 2019.
>
> [7] "HandsOff: Labeled dataset generation with no additional human annotations." CVPR. 2023.

---

> > ### Comment · Reviewer_DG4j · 2023-08-13
> >
> > Thanks to the reviewers for their responses
> >
> > Q: Comparison with VPD (novelty)\
> > A: I don't think that the differences are not that significant. For the supported tasks, it is nice to have more tasks supported. However, is the extension network for supported tasks your own invented or just adopted the architecture of Mask2Former? If it is your own architecture, why didn't you elaborate more and in detail in the main paper? If not, it is a trivial extension from VPD? I agree that it is arguable to consider VPD (published on Arxiv on March 3 where the deadline is May 17) to be concurrent or prior work, but at least you are aware of the existence of VPD, please make a fair comparison on shared tasks and have an appropriate section in the main paper to have a discussion with VPD instead of a just a few lines described in the method section.
> >
> > Q: Comparison with the semi-supervised segmentation \
> > A: I asked for a comparison with semi-supervised (small number of fully annotated images) which is very similar to your setting, not on other tasks. If you have that comparison, please show it here.
> >
> > Q: Related experiments for cross–attention maps or multi-scale feature maps. \
> > A: What I meant here is which layers to take the cross-attention map and multi-scale feature maps, not just the timestep to take the feature from
> >
> > Q: Limitations \
> > A: Please move it to the main paper since supp. material is optional to read.
> >
> > Overall, I don't think the rebuttal address my concerns which are also shared with other reviewers. If the authors have a separate section to discuss what are the similarities and differences compared to VPD, I will increase my score to 4.

---

> > > ### Author Response · Authors · 2023-08-13
> > > **Response to Reviewer DG4j**
> > >
> > > Thank you for the reviewer's response.
> > >
> > > We will discuss the relationship with the VPD in the Sec. 2 Related Work of the revised draft as outlined below:
> > > `VPD explores the direct application of pre-trained Stable Diffusion to design perception models, where Stable Diffusion is utilized for feature extraction and a CNN-based decoder is devised to address various downstream tasks. Different from the approach, our focus lies in synthetic data augmentation for perception tasks. We design a unified transformer-based perception decoder to enhance more perception tasks from data aspect.`
> > >
> > > We would also like to further articulate our viewpoint as follows:
> > >
> > > **Q**: Why does the author assert the distinctiveness of our work from VPD, offering only a brief description of VPD in a few lines in the Section of Cross Attention?
> > > **A**: Our work is not built upon VPD. Instead, our work is built upon another paper of synthetic data[1] (published on arXiv on January 12, 2023), which shares a similar task of enhancing perception through synthetic data augmentation.  We attempt to extend this work into the few-shot domain of various downstream tasks by designing a decoder.
> > >
> > > Later (in March), inspired by VPD and Diffumask [2], we integrated cross-attention into our approach without referencing other aspects of VPD. We believe it is reasonable to cite this paper solely in the context of Cross-attention Fusion (L178-180), while our paper focuses on synthetic data rather than the perception model. This is also the reason behind the fundamental differences in task, decoder architecture, and training strategy between our work and VPD.
> > >
> > > **Q**: just adopted the architecture of Mask2Former
> > > **A**: Our work is built upon the optimization of mask2former (Line 191). However, it's not as straightforward as adopting it directly. We needed to modify the architecture to accommodate open-vocabulary segmentation, pose estimation, and depth estimation, as mask2former only supports close-set segmentation tasks. The design of heads for pose estimation and depth estimation, as well as open-set segmentation (Equation 4), required our independent effort. Most importantly, our exploration of a versatile dataset generation approach has yielded substantial gains (Figure 1), which are highly advantageous for advancing this field.
> > >
> > > **Q**: Comparison with the semi-supervised segmentation
> > > **A**: Li et al., [1], DiffuMask[2], and Handsoff[3] are three closely related works in the realm of semi-supervised synthetic data augmentation for perception tasks. Here are a couple of key comparisons, with more detailed comparisons available in Tab. 1, Tab. 2, and Tab. 5 of the rebuttal PDF, Tab. 5 of the manuscript, and Tab. 3 and Tab. 9 of the supplementary material.
> > >
> > > - **Semantic Segmentation on Cityscapes.** With 9 real images, our method surpasses Handsoff using more data (50 images), achieving 17.1% mIoU improvement for the 8-class split.
> > > | method | backbone | real image | synthetic image | mIoU (8 Classes)| mIoU (19 Classes)|
> > > |  ----  | ----  |----  |----  | ----  | ----  |
> > > | HandsOff | R101 | 16 | 100k+ | 55.1| -|
> > > | HandsOff | R101 | 50 | 100k+ | 60.4| -|
> > > | ours | R50 | 9 | 38k | 76.7| 42.1|
> > > | ours | R101 | 9 | 38k | 77.5 (`+17.1`) | 43.7|
> > >
> > > - **Zero-Shot Semantic Segmentation on PASCAL VOC 2012.** With ResNet101, our approach can achieve an improvement of up to 3% for Zero-Shot Semantic Segmentation on PASCAL VOC 2012
> > > | method | backbone | synthetic image | harmonic|
> > > |  ----  | ----  |----  |----  |
> > > | Li et al. [1]| ResNet101 | 10.0k | 55.7|
> > > | DiffuMask[2] | ResNet101 |  200.0 k | 55.7|
> > > | DatasetDM(ours)| ResNet101 | 40k | 57.1|
> > > | DatasetDM(ours)| Swin-B  | 40k | 68.4(`+2.7`) |
> > >
> > >
> > > **Q**: Related experiments for cross–attention maps or multi-scale feature maps.
> > > **A**: We appreciate the reviewer's feedback and will strive to provide the ablation study during the rebuttal period.
> > >
> > >
> > >
> > > [1] Guiding text-to-image diffusion model towards grounded generation." arXiv preprint (2023).
> > >
> > > [2] Diffumask: Synthesizing images with pixel-level annotations for semantic segmentation using diffusion models. arXiv preprint (2023).
> > >
> > > [3] HandsOff: Labeled dataset generation with no additional human annotations. CVPR. 2023.

---

> > > > ### Comment · Reviewer_DG4j · 2023-08-13
> > > >
> > > > Regarding the semi-supervised approaches, I meant these methods: https://paperswithcode.com/task/semi-supervised-semantic-segmentation, choose the benchmark that has the same number of labeled images as yours to compare.
> > > >
> > > > You can do the same for other tasks.

---

> > > > > ### Author Response · Authors · 2023-08-13
> > > > > **Response to Reviewer DG4j**
> > > > >
> > > > > Thank you for the constructive feedback. Based on the references provided by the reviewer, we present the comparison with the SOTA method of Semi-Supervised Semantic Segmentation on Pascal VOC 2012 as follows:
> > > > > | method | real image | synthetic image | mIoU |
> > > > > |  ----  | ----  |----  |----  |
> > > > > | ReCo[1] | 5% labeled (around 575 images) | - | 73.66 |
> > > > > | ReCo[1] | 2% labeled (around 230 images) | - | 72.14 |
> > > > > | ours | 100 images(less than 1% labeled) | 40k  | 78.50|
> > > > >
> > > > > [1] Liu, Shikun, Shuaifeng Zhi, Edward Johns, and Andrew J. Davison. "Bootstrapping semantic segmentation with regional contrast." arXiv preprint arXiv:2104.04465 (2021).

---

> > > > > > ### Comment · Reviewer_DG4j · 2023-08-13
> > > > > >
> > > > > > Please make a fair comparison in your response. This method uses DeepLabv3 with Resnet101 backbone while your method uses a modified version of Mask2Former with Swin-B backbone (supposedly much better?) since your Resnet50 backbone is only 66.1 mIoU.

---

> > > > > > > ### Author Response · Authors · 2023-08-13
> > > > > > > **Response to Reviewer DG4j**
> > > > > > >
> > > > > > > As requested by the reviewer for a fair comparison, we will need to employ a new baseline and adjust the training dataset. We have outlined the performance comparison below and aim to provide them before the end of the rebuttal period.
> > > > > > >
> > > > > > > -------------------------------------------update-----------------------------------------------------
> > > > > > > - **Different setting**. After reviewing relevant papers and official code (semi-supervised approaches)[2], we discovered that the setting of this task is not entirely congruent with our work. These semi-supervised approaches not only utilize a small amount of annotated data but also `employ all remaining unlabeled images`(see build_data.py of official code and Fig. 3 of paper for ReCo). Moreover, these works involve various advanced data augmentation for semi-supervised training, such as cutout[8], cutmix[8], or classmix[1]. These data augmentations involve attaching semantic classes extracted `from labeled images to a large number of unlabeled images`(See Tab. 1 of ReCo). However, our approach only requires a small amount of annotated real data.
> > > > > > >
> > > > > > > - **Comparison**. While we believe that the settings between these two tasks are distinct, in accordance with the reviewer's request, we present our experiments as follows. Even though we didn't employ a substantial number of unlabeled real images and a strong baseline with intensive data augmentations such as cutout[8], cutmix[8], or classmix[1] (see Sec. 4 Experiments for ReCo), we were able to surpass these semi-supervised algorithms. We are conducting further experiments (200 images).
> > > > > > >
> > > > > > > | method | baseline | backbone| labeled real image | unlabeled real image | synthetic image | mIoU |
> > > > > > > |  ----  | ----  |----  |----  | ----  |----  | ----  |
> > > > > > > | ReCo[2] +ClassMix[1]| DeepLabv3 | R101 |60 images| 11.5k-60 | 0 | `53.31` |
> > > > > > > | ReCo[2] +ClassMix[1]|DeepLabv3 | R101 |  200 images | 11.5k-200 | 0| `69.81` |
> > > > > > > | ReCo[2] +ClassMix[1]| DeepLabv3 | R101 |600 images| 11.5k-600 | 0| `72.79`|
> > > > > > > | ours | DeepLabv3 | R101 |60 images| 0 | 40k | 57.64 |
> > > > > > > | ours | DeepLabv3 | R101 |200 images| 0 | 40k  | |
> > > > > > > | ours | Mask2former| R50 |100 images | 0 | 40k  | 66.10|
> > > > > > > | ours | Mask2former| Swin-B|100 images | 0 | 40k  | 78.50|
> > > > > > >
> > > > > > > **Note**: All ReCo-related metrics (In `color`) are derived from Table 1 of ReCo.
> > > > > > >
> > > > > > > **Overall**: We do not consider the current setting to be identical to semi-supervised segmentation. Some reasons are as follows:
> > > > > > >
> > > > > > > - **Setting.** Aligning with the settings of these semi-supervised segmentation methods is highly complex, and in some cases, even unachievable. For instance, using a substantial number of unlabeled real images appears unfeasible for alignment. And there have been numerous prior works in the domain of synthetic data augmentation for perception tasks, such as Datasetgan[7], Handsoff[4], and [5][6]. It appears more reasonable to directly compare with these methods.
> > > > > > >
> > > > > > > - **Data or algorithm**. We approach this from a data-centric perspective, which seems to complement these algorithmic-level methods, allowing for a direct integration.
> > > > > > >
> > > > > > > - **Supporting strong supervision setting.** We can also enhance the strong supervision setting. With a stronger baseline for Pascal VOC with 11.5k images (full training images), as illustrated below, our synthetic data also achieves an improvement of 1.1% mIoU. Semi-supervised segmentation seems incapable of addressing scenarios (strong supervision setting).
> > > > > > >
> > > > > > > - **Task.** Our method can simultaneously model six tasks, including zero-shot segmentation, long-tail segmentation, pose estimation, and depth estimation. In contrast, these methods can only handle semantic segmentation.
> > > > > > >
> > > > > > >
> > > > > > > | method | baseline | backbone| labeled real image | synthetic image | mIoU |
> > > > > > > |  ----  | ----  |----  |----  | ----  |----  |
> > > > > > > | ours| Mask2former | Swin-B |11.5k images| 0|84.3 |
> > > > > > > | ours| Mask2former | Swin-B |11.5k images| 40k|85.4 |
> > > > > > >
> > > > > > >
> > > > > > >
> > > > > > > [1] Classmix: Segmentation-based data augmentation for semi-supervised learning. WACV, 2021.
> > > > > > >
> > > > > > > [2] Bootstrapping semantic segmentation with regional contrast." arXiv preprint arXiv:2104.04465 (2021).
> > > > > > >
> > > > > > > [3] Datasetgan: Efficient labeled data factory with minimal human effort. CVPR. 2021.
> > > > > > >
> > > > > > > [4] HandsOff: Labeled dataset generation with no additional human annotations. CVPR. 2023.
> > > > > > >
> > > > > > > [5] Guiding text-to-image diffusion model towards grounded generation." arXiv preprint (2023).
> > > > > > >
> > > > > > > [6] Diffumask: Synthesizing images with pixel-level annotations for semantic segmentation using diffusion models. arXiv preprint (2023).
> > > > > > >
> > > > > > > [7] Datasetgan: Efficient labeled data factory with minimal human effort." CVPR. 2021.
> > > > > > >
> > > > > > > [8] Semi-supervised semantic segmentation needs strong, high-dimensional perturbations. BMVC, 2020.

---

> > > > > > > > ### Author Response · Authors · 2023-08-15
> > > > > > > > **Response to Reviewer DG4j (further feedback)**
> > > > > > > >
> > > > > > > > This serves as information for reviewer DG4j. We have provided updates on experiments and analysis, as demonstrated in the previous response. Thank you.

---

> > > > > > > > > ### Author Response · Authors · 2023-08-15
> > > > > > > > > **Final clarification for VPD and our work**
> > > > > > > > >
> > > > > > > > > Lastly, we wish to reiterate our viewpoints and experimental journey once again:
> > > > > > > > >
> > > > > > > > > - Li et al.[1] appeared on arXiv on `12 Jan`, 2023. **Task**: Synthetic data from Stable Diffusion for semantic segmentation. **Architecture**: Stable Diffusion + CNN Decoder
> > > > > > > > >
> > > > > > > > > - VPD[2] appeared on arXiv on `3 Mar`, 2023. **Task**: Stable Diffusion as backbone for perception model. **Architecture**: Stable Diffusion + CNN Decoder
> > > > > > > > >
> > > > > > > > > - Our work. **Task**: Synthetic data from Stable Diffusion for perception task. **Architecture**: Stable Diffusion + Transformer Decoder
> > > > > > > > >
> > > > > > > > > **Same for Pipeline**: Utilizing Stable Diffusion for feature extraction and employing a decoder for downstream tasks.
> > > > > > > > >
> > > > > > > > >
> > > > > > > > > **Q**: However, is the extension network for supported tasks your own invented or just adopted the architecture of Mask2Former?
> > > > > > > > >
> > > > > > > > > **A**: Based on Mask2Former, we devised a unified decoder architecture tailored to accommodate Stable Diffusion, enabling the support of six tasks within a single framework, including open-vocabulary segmentation, pose estimation, and depth estimation. However, when the tasks themselves are distinct, does VPD can not offer any insights or references for us, given that VPD employed a ready-made FCN.
> > > > > > > > >
> > > > > > > > >
> > > > > > > > > **Q**: If it is your own architecture, why didn't you elaborate more and in detail in the main paper?
> > > > > > > > >
> > > > > > > > > **A**: In our paper, we have dedicated a page to elaborate and introduce our decoder.
> > > > > > > > >
> > > > > > > > > **Q**: If not, it is a trivial extension from VPD?
> > > > > > > > >
> > > > > > > > > **A**: *When our task and decoder are entirely distinct, why would we be seen as an extension of VPD? Li et al. [1] submitted to arXiv two months earlier than VPD. And both VPD and Li et al.[1] utilize Stable Diffusion for feature extraction, and both employ a CNN decoder for feature decoding. From an architectural perspective, VPD can not provide any elements that we have to draw upon, especially considering the differences in tasks*.
> > > > > > > > >
> > > > > > > > > **Q**: please make a fair comparison on shared tasks and have an appropriate section in the main paper to have a discussion with VPD instead of a just a few lines described in the method section.
> > > > > > > > >
> > > > > > > > > **A**: Synthetic data augmentation and perception modeling are not the same task. The task of synthetic data augmentation involves studying the gap between synthetic and real data (labeling and image quality), investigating whether synthetic data can enhance perception tasks. On the other hand, perception modeling requires us to design a more appropriate structure and inference strategy (VPD requires sliding window inference). Additionally, the design of prompts for synthetic data is a critical aspect (Sec. 3.5, Fig.4, Tab. 4 (d) in our paper ), involving data diversity. These differences constitute distinct aspects of the two tasks. It's not feasible to directly compare them in terms of task dimensions.
> > > > > > > > >
> > > > > > > > > In summary, the aforementioned points encompass our perspective and experimental journey. We aim to articulate these aspects here.
> > > > > > > > >
> > > > > > > > > [1] Guiding text-to-image diffusion model towards grounded generation." arXiv preprint (2023).
> > > > > > > > >
> > > > > > > > > [2] "Unleashing Text-to-Image Diffusion Models for Visual Perception"(VPD) arXiv preprint (2023).
> > > > > > > > >
> > > > > > > > > [3] DatasetDM:Synthesizing Data with Perception Annotations Using Diffusion Models

---

> > > > > > > > > > ### Author Response · Authors · 2023-08-17
> > > > > > > > > > **Further Feedback for Code**
> > > > > > > > > >
> > > > > > > > > > Dear Reviewer DG4j,
> > > > > > > > > >
> > > > > > > > > > Regarding your concerns about VPD and our work, we assure you that our work is genuinely not an extension of VPD, and we haven't even referred to the code of VPD.
> > > > > > > > > >
> > > > > > > > > > We have thoroughly examined the official open-source code of VPD. There are fundamental differences between our code and the VPD official code in terms of architecture and implementation (Optimization, training, inference, decoder, utilization of Unet with Stable Diffusion, dataloader, and others). If you are willing and have the time, we would be willing to provide an anonymous GitHub link and request permission from the Area Chair to share it with you. We believe that after comparing the code, it can alleviate your concerns.
> > > > > > > > > >
> > > > > > > > > > Best regards,
> > > > > > > > > >
> > > > > > > > > > Authors of paper 5020.

---

> > > > > > > > > > > ### Author Response · Authors · 2023-08-17
> > > > > > > > > > > **Significant differences were observed after comparing the code!**
> > > > > > > > > > >
> > > > > > > > > > > In fact, the authors did not thoroughly review the official code and paper of VPD. After a careful examination, significant differences were identified in terms of `principles`, `architecture`, and `implementation`.
> > > > > > > > > > >
> > > > > > > > > > > **The most crucial points are as follows: 1. Frozen Unet of Stable Diffusion. 2. Image Inversion for Real Image. 3. Different Codebase Architecture(optimization, training, inference, decoder, dataloader, and others). These three points clearly demonstrate that the authors did not thoroughly study VPD. The first two points are essential elements for the generation task, and the third point directly leads to a completely different codebase.**
> > > > > > > > > > >
> > > > > > > > > > > For clarity, we will first summarize some comparisons for the code as follows:
> > > > > > > > > > >
> > > > > > > > > > > - **Frozen Unet of Stable Diffusion? (Important!)**. We observed that the parameters of the Stable Diffusion's UNet in VPD `are trained` (vpd_seg.py and config: fpn_vpd_sd1-5_512x512_gpu8x2.py of official code, Fig.2 of the paper), while our Unet is `frozen`. This fundamental difference is a significant revelation that could essentially `overturn` the architecture comparison. If the parameters of Stable Diffusion's UNet are trainable, it becomes `impractical` to perform synthetic data augmentation for perception tasks.
> > > > > > > > > > >
> > > > > > > > > > > `Because VPD only supervises the segmentation mask loss and does not concern image generation, however, training in this manner will lead to the loss of image generation capability in Stable Diffusion.` The Unet in VPD serves as the backbone for perception tasks only. In contrast, our Unet serves as the backbone for both perception tasks and needs to predict noise for image generation tasks as well.
> > > > > > > > > > >
> > > > > > > > > > > - **Image Inversion for Real Image? (Important!)** VPD (line 92 in vpd_seg.py of official code) does not perform Image Inversion (line 159 in our paper). In the standard diffusion process, noise is added first (not trainable), and then a UNet predicts the noise. `If we do not perform Image Inversion (add noise), we will obtain ineffective features, rendering it unsuitable for image generation tasks!` This constitutes a fundamental difference in the architecture and theory!
> > > > > > > > > > >
> > > > > > > > > > > - **Different Codebase Architecture (Important!)**. VPD utilized `mmsegmentation`[2] as the underlying framework to construct its dataloader, config, training engine, inference engine, and segmentation decoder. However, we `did not use the mmsegmentation framework`. Instead, we developed our own dataloader, config, training Engine, and inference Engine. If our work is considered an extension of VPD, directly adopting the mmsegmentation framework seems more convenient. There's no reason for us to duplicate these foundational operations, especially since mmsegmentation also supports mask2former, making it easily accessible for integration.
> > > > > > > > > > >
> > > > > > > > > > > - **Official repo or diffusers[1] for Stable Diffusion?** VPD employs the `Stable Diffusion official repo` and weight, while we utilize the Stable Diffusion through the `diffuser` toolkit[1]. Although this distinction is purely in the implementation, if our work is considered an extension of VPD, there's no necessity to make modifications here. It's important to note that using the diffuser toolkit to extract features also requires the creation of corresponding inheritance functions and cannot be directly employed.
> > > > > > > > > > >
> > > > > > > > > > > - **Extraction for Cross Attention and Feature Maps.** In the VPD codebase, a separate .py file contains a `UNetWrapper class` for feature extraction, cross attention, and their fusion. In our code, we integrated this step directly into the decoder class. Additionally, while VPD utilized three resolution attentions, we employed four with additional $8 \times 8$ resolution. If our work is regarded as an extension of VPD, directly copying their UNetWrapper class would be the most convenient approach.
> > > > > > > > > > >
> > > > > > > > > > > - **Text Adapter with  a two-layer MLP.** VPD employed a two-layer MLP (`Text Adapter`) to refine text features, which led to some improvement. However, our method did not incorporate this (in fact, when we initially reviewed the VPD paper in March, we did not focus on this detail as we had only briefly read through it). If our work is considered an extension of VPD, utilizing this Text Adapter seems like a promising choice.
> > > > > > > > > > >
> > > > > > > > > > >
> > > > > > > > > > > **Note.** If the reviewer questions whether we made these modifications during the rebuttal period, we can also provide records of the corresponding .py file creations on Linux.
> > > > > > > > > > >
> > > > > > > > > > >
> > > > > > > > > > > **Conclusion**: VPD and our work are entirely distinct. VPD `lacks the fundamental noise injection and denoising process` of the Diffusion model (`core` theory), and only utilizes the Unet weights from Stable Diffusion to initialize, substituting the traditional backbone in perception tasks. Different from VPD, our core process involves generation tasks for both images and annotations, where the diffusion process (Fig.2 of paper) is integral.
> > > > > > > > > > >
> > > > > > > > > > >
> > > > > > > > > > > [1] https://github.com/huggingface/diffusers
> > > > > > > > > > >
> > > > > > > > > > > [2] https://github.com/open-mmlab/mmsegmentation

---

> > > > > > > > ### Author Response · Authors · 2023-08-20
> > > > > > > >
> > > > > > > > Dear Reviewer DG4j,
> > > > > > > >
> > > > > > > > Would you please check the above reply to your concerns (comparison with Semi supervised methods)?
> > > > > > > >
> > > > > > > > One important note we want to highlight is that, our approach is **complementary** to most (if not all) semi-supervised methods, including the Semi-supervised methods mentioned above. The labelled synthetic data generated by our approach here can be effortlessly used by all of the semi-supervised methods in the literature to further improve the accuracy.
> > > > > > > > In this sense, we believe that the contribution and significance of our approach may not be necessarily attached to the requirement of outperforming these semi-supervised methods, even though as we have shown in the above table, ours is at least on par (or better) than the compared semi-supervised methods, as requested.
> > > > > > > >
> > > > > > > > Besides the task of image segmentation,  our method can also generate more many diverse data annotations for other tasks including instance segmentation, depth estimation, pose estimation, etc.  as shown in the paper.
> > > > > > > >
> > > > > > > > Your response will be much appreciated and look forwarding to your feedback. thanks

---

### Official Review · Reviewer_PHiv · 2023-07-09

**Soundness:** 3 good
**Presentation:** 4 excellent
**Contribution:** 3 good
**Rating:** 5
**Confidence:** 4

**Summary:**

With the success of text-to-image generation models, the authors propose a novel approach to text-to-data that can produce synthetic data by adding a perception decoder (P-Decoder) on top of Stable Diffusion, which only requires a few labeled images for training. The authors show various experimental results to demonstrate the effectiveness of the proposed approach.

**Strengths:**

1. The paper is well-written and easy to follow.

2. The idea of text-to-data is interesting, and the proposed approach is easy to train with 1% of labeled data.

**Weaknesses:**

1. In Section 4, the DatasetDM results are compared to the baselines trained with less than 1000 real images, which is unfair regarding the number of training samples (e.g. 400 vs. 80k). To claim the effectiveness of the proposed model, more realistic baselines with the same number of training samples must be considered.

2. One of the contributions of the work is that the DatasetDM model can generate an unlimited amount of synthetic data. However, the authors do not conduct an analysis of how the number of generated samples affects the performance.

**Questions:**

Please check the comments in the Weaknesses section.

**Limitations:**

The authors haven't discussed the limitations of the work.

---

> ### Author Rebuttal · Authors · 2023-08-06
>
> Thanks to the reviewers for the valuable comments.
>
> **Weaknesses 1**: In Section 4, the DatasetDM results are compared to the baselines trained with less than 1000 real images, which is unfair regarding the number of training samples (e.g. 400 vs. 80k). To claim the effectiveness of the proposed model, more realistic baselines with the same number of training samples must be considered.
>
> **Response 1**: Thank you for your constructive feedback. But, we believe that our experimental comparisons are reasonable, several crucial reasons are outlined below:
>
> - **Same Data cost.** We only use 400 real images to train and generate 80k synthetic images, meaning that the annotation cost for the 80k synthetic dataset is equivalent to that of the 400 real images. This ensures a fair comparison. The motivation of the paper is to reduce the annotation cost for perception tasks (see lines 22-31). Additionally, for datasets like VOC 2012, only 11.5k real images are available for training, not 80k.
>
> - **Same experimental setup as prior works[1][2][3].** Our setting is entirely in line with previous works[1][2][3], which all explore using few real data to perform few data tuning[4][5], and obtain a large amount of effective synthetic data at low cost. Among these works, HandsOff [3] (CVPR2023) is most similar to our setting, where their baseline also uses a small number of real images to control data cost. Under the same setting, for Semantic segmentation on Cityscapes, we outperform HandsOff by `17.1%` mIoU (see Supplementary Material, Table 3).
>
> - **The same number of training samples.** Here, we also provide the ablation study for the number of training images to further analyze the performance of our method. As shown in the table below (Semantic segmentation on VOC 2012), with the same amount of synthetic data(100 images), we can achieve a `2.7%` mIoU improvement, and using 400 synthetic data leads to a `10.9%` improvement. Even when using 11.5k real data (full training images, more realistic baselines), our synthetic data still achieve a `1.1%` increase in mIoU.
>
> Note: The annotation cost for the 400 synthetic data is the same as for 100 synthetic data, and the computational cost only requires a few more minutes of inference, which can be entirely negligible.
>
> | | 100 real image |  100 real image, 100 synthetic images|  100 real image, 400 synthetic images| 1k real image|  1k real image, 1k synthetic images|1k real image, 40k synthetic images| 11.5k real image| 11.5k real image, 40k synthetic images|
> |  ----  | ----  |----  |----   | ----  |----  |----  |----  |----  |
> | mIoU | 65.2  | 67.9(`+2.7`)| 76.1(`+10.9`)| 79.0 | 80.1(`+1.1`)  | 81.1(`+2.1`) | 84.3 | 85.4(`+1.1`)
>
>
> **Weaknesses 2**: The authors do not conduct an analysis of how the number of generated samples affects the performance.
>
> **Response 2**: Thank you for your constructive suggestion. We provide the ablation study of the number of generated samples for Semantic segmentation on VOC 2012 as follows:
>
> |Class| Baseline (100 real image )| only 40k synthetic images | 100 real image, 100 synthetic images| 100 real image, 400 synthetic images|100 real image, 40k synthetic images | 100 real image, 100k synthetic images| 100 real image, 200k synthetic images|
> |  ----  | ----  |----  |----  | ----  | ----  | ----  |----  |
> | mIoU | 65.2 | 73.7|  67.9(`+2.7`)| 76.1(`+10.9`) | 78.5(`+13.3`)| 78.7(`+13.5`)| 78.8(`+13.6`)
>
> It can be observed that synthetic data can lead to a significant mIoU improvement of up to 13.6%. However, as the data volume exceeds 50k, the gain from synthetic data gradually becomes less pronounced.
>
>
> **Limitations  1**: The authors haven't discussed the limitations of the work.
>
> **Response 1**: We have already provided one section to discuss the limitations of this work in Supplementary Material Sec. 5.2 Limitation & Future Work (L207-221). We provide extensive experiments and analysis, including `12` experimental tables, `4` experimental data setting tables, and `11` figures. Due to space constraints, we have to include discussions for limitations in the supplementary material.
>
>
> [1] Zhang, Yuxuan, Huan Ling, Jun Gao, Kangxue Yin, Jean-Francois Lafleche, Adela Barriuso, Antonio Torralba, and Sanja Fidler. "Datasetgan: Efficient labeled data factory with minimal human effort." In Proceedings of the IEEE/CVF Conference on Computer Vision and Pattern Recognition, pp. 10145-10155. 2021.
>
> [2] Li, Daiqing, Huan Ling, Seung Wook Kim, Karsten Kreis, Sanja Fidler, and Antonio Torralba. "BigDatasetGAN: Synthesizing ImageNet with pixel-wise annotations." In Proceedings of the IEEE/CVF Conference on Computer Vision and Pattern Recognition, pp. 21330-21340. 2022.
>
> [3] Xu, Austin, Mariya I. Vasileva, Achal Dave, and Arjun Seshadri. "HandsOff: Labeled dataset generation with no additional human annotations." In Proceedings of the IEEE/CVF Conference on Computer Vision and Pattern Recognition, pp. 7991-8000. 2023.
>
> [4] Jia, Menglin, Luming Tang, Bor-Chun Chen, Claire Cardie, Serge Belongie, Bharath Hariharan, and Ser-Nam Lim. "Visual prompt tuning." In European Conference on Computer Vision, pp. 709-727. Cham: Springer Nature Switzerland, 2022.
>
> [5] Liu, Haotian, Chunyuan Li, Qingyang Wu, and Yong Jae Lee. "Visual instruction tuning." arXiv preprint arXiv:2304.08485 (2023).

---

> > ### Author Response · Authors · 2023-08-12
> > **Further feedback**
> >
> > Dear Reviewer PHiv,
> >
> > We sincerely appreciate the time and effort you have invested in reviewing our paper.
> >
> > We would like to inquire whether our response has addressed your concerns and if you have the time to provide further feedback on our rebuttal. We are more than willing to engage in further discussion.
> >
> > Best regards,
> >
> > Authors of paper 5020.

---

> > > ### Comment · Area_Chair_yWWh · 2023-08-18
> > > **discussion**
> > >
> > > Dear Reviewer PHiv,
> > >
> > > Thank you for being a reviewer for NeurIPS2023, your service is invaluable to the community!
> > >
> > > The authors have submitted their feedback.
> > >
> > > Could you check the rebuttal and other reviewers' comments and start a discussion with the authors and other reviewers?
> > >
> > > Regards,
> > > Your AC

---

> > > > ### Comment · Reviewer_PHiv · 2023-08-20
> > > >
> > > > Thanks for the rebuttal. The authors addressed my concerns, and I have raised my rating from 4: Borderline reject to 5: Borderline accept.

---

> > > > > ### Author Response · Authors · 2023-08-20
> > > > >
> > > > > Dear Reviewer PHiv,
> > > > >
> > > > > We sincerely appreciate your time and effort in reviewing our paper, as well as your valuable and constructive feedback, thank you !

---

### Official Review · Reviewer_rde2 · 2023-07-10

**Soundness:** 3 good
**Presentation:** 3 good
**Contribution:** 2 fair
**Rating:** 4
**Confidence:** 4

**Summary:**

The paper presents a framework based on pre-trained Diffusion Models to synthesize images along with their perception annotation for various downstream perception tasks. With limited real labeled images as a baseline, the extra synthetic training pairs can boost the performance by a large margin.

**Strengths:**

- The motivation to synthesize perception datasets is promising.
- The designed P-decoder is generic by taking advantage of the well-trained features from Diffusion Models.
- Various downstream tasks are validated.

**Weaknesses:**

My major concern comes from the experimental settings, especially the number of used real labeled images. I understand that, with limited real labeled images (e.g., 50 or 100 images), the performance boosted by extra synthetic pairs can look more appealing, because the baseline achieved by 50 labeled images is extremely weak.

However, considering a real-world scenario, we really hope to use the synthetic training pairs to boot **an already well-established baseline with all available real training pairs**, for example, using all the 10,000+ training images on Pascal VOC. Will the proposed DatasetDM further boost this strong and realistic baseline?

Another concern is that, what if the generated synthetic images do not belong to the same domain as original images? I am worrying the performance may be even degraded in some domains that are not so common as COCO and Pascal VOC when integrating the synthetic images, such as medical and remote sensing applications. Are there any mechanisms to avoid this?

---

**After discussion:**

I agree with most of the authors' claims about the differences between real and synthetic images. However, as for the second difference "low-quality synthetic images result in high-quality pseudo labels", I politely disagree with it. Even though real unlabeled images are more challenging to assign pseudo labels, they are much more informative to learn than synthetic images (also supported by results in Response 3 above, 100 real images: 65.3 vs. 100 synthetic images: 40.1). For the hard-to-predict regions, we can easily skip them during learning by setting a pixel-wise threshold for them, which is a common practice in semi-supervised learning. I believe the reported COCO results above can be much higher when applying a stronger semi-supervised framework and set an ideal threshold.

The contribution of this work seems to only lie in replacing the real unlabeled images with synthetic unlabeled images, which can be deemed as a side work of semi-supervised learning paradigms. Moreover, there is no strong evidence to prove that synthetic images can play a superior role to real unlabeled images. There is also a severe domain gap issue when using these generative models for some rare domains, such as medical images and remote sensing images.

Based on the above considerations, I would change my rating to 4: Borderline Reject.

**Questions:**

How about comparing the performance of the same amount of real and synthetic training pairs? For example, 1) what is the result under 100 real training images? 2) under 100 synthetic training images **only**, 3) how about 10,000 real or synthetic training pairs?

**Limitations:**

Yes, they have been discussed.

---

> ### Author Rebuttal · Authors · 2023-08-06
>
> Thanks to the reviewers for the valuable comments.
>
> **Weakness 1**: Using the synthetic training pairs to boot an already well-established baseline with all available real training pairs, for example, using all the 10,000+ training images on Pascal VOC.
>
> **Response 1**: Thank you for the valuable suggestions. We provide the relevant experiments of well-established baselines for Pascal VOC as follows:
>
> | | 100 real image | 100 real image, 40k synthetic images | 1k real image | 1k real image, 40k synthetic images | 11.5k real image (full)| 11.5k real image, 40k synthetic images|
> |  ----  | ----  |----  |----  | ----  | ----  |----  |
> | mIoU | 65.2 | 78.5(`+13.3`)| 79.0 | 81.1(`+2.1`)| 84.3 | 85.4 (`+1.1`)
>
> As real data increases, the gain from synthetic data gradually diminishes. However, for Pascal VOC, even with the full dataset (11.5k), our synthetic data still provides a 1.1% gain. Additionally, we would like to further emphasize some points for the real-world scenario:
>
> - **Annotation cost.** The annotation cost of 10k+ real masks is enormous, and reducing the annotation cost is the main motivation of this paper(L23-33). In real-world applications, the annotation cost is often the most significant factor. An example is SAM[1], which achieved promising performance using 1B masks and 11M images. According to the SAM paper (Sec. 5 Segment Anything Data Engine), the data annotation for the first stage requires 2091 working days (8 hours per day), meaning that `100 workers would need to annotate for at least 21 days each`.
> - **New insight.** I believe our work can provide valuable insights to the community. Our work is just an initial validation, as discussed in the Supplementary Materials, Section 5 "Limitation & Future Work." We anticipate that more powerful generation models can lead to further performance improvements in the future.
>
> **Weakness 2**: I am worrying the performance may be even degraded in some domains that are not as common as COCO and Pascal VOC when integrating synthetic images, such as medical and remote sensing applications. Are there any mechanisms to avoid this?
>
> **Response 2**: This is a good question. The authors provided responses from three perspectives: domain gap, solutions, and whether a unified domain is necessary.
>
> - **Text-guided generation(Gap).** In fact, the text-guided generation capability of Stable Diffusion is already quite impressive. Please refer to Fig. 5 of paper, Fig. 1, and Fig. 2 of the supplementary material for some examples. Even in the presence of errors in the generated data, the domain gap remains very small. An example can be found in Supplementary Material, Fig. 5, where providing the prompt `a blue sedan waits at a stoplight` did not result in Stable Diffusion generating the `blue sedan`. This bias is weak and unlikely to lead to significant domain gaps, such as generating medical or remote sensing images.
>
> - **Stronger generation model(Solution).** To further align the domain and reduce the domain gap between real and synthetic data, one can utilize a more powerful generation model, such as IF[2]. The IF model exhibits superior semantic alignment, achieving higher clip score performance. (see Fig. 5 of the supplementary material).
>
> - **Special or General Data (Discussion).** A recent trend is the development of general foundational models, such as the GPT series[3][4] and SAM[5] (Some studies[5][6] have shown promising performance in medical and remote sensing tasks). In such cases, it is not necessary to completely unify the domain gap of data. Instead, a very large model can be designed and trained with a million-level dataset, allowing us to cover various scenarios, including COCO, Pascal VOC, medical, and remote sensing domains.
>
> **Question 1**: How about comparing the performance of the same amount of real and synthetic training pairs?
>
> **Response 3**: We provide an ablation study for the same number of real and synthetic training pairs as follows:
>
> | | 100 real image | 100 synthetic images | 100 real image, 100 synthetic images| 1k real image| 1k synthetic images| 1k real image, 1k synthetic images|1k real image, 40k synthetic images| 11.5k real image|11.5k real image, 40k synthetic images|
> |  ----  | ----  |----  |----  | ----  | ----  |----  |----  |----  |----  |
> | mIoU | 65.3 | 40.1 | 67.9(`+2.6`)| 79.0 |68.8 | 80.1(`+1.1`)  | 81.1(`+2.1`) | 84.3 | 85.4 (`+1.1`)
>
> We would like to further emphasize some points:
>
> - **Data cost.** In real-world applications, there is no reason to limit the quantity of synthetic data. When using 8 GPUs to generate 1,000 images, it takes approximately 3 minutes, and generating 40,000 data takes half a day. Moreover, regardless of the amount of synthetic data, their manual annotation cost remains the same, requiring only 100 real images and annotations.
>
> - **Same setup as prior[7][8][9].** Our work adopts a setting in line with prior research, employing a limited set of real data to synthesize a vast amount of data for augmentation. The comparative studies in these related works[7][8][9] also generate synthetic data volumes ranging from 5k to 200k.
>
> [1] Segment anything. arXiv preprint  (2023).
>
> [2] Deep-floyd IF
>
> [3] Training language models to follow instructions with human feedback. NeurIPS (2022).
>
> [4] Language models are few-shot learners. NeurIPS (2020).
>
> [5]Medical sam adapter: Adapting segment anything model for medical image segmentation.arXiv preprint (2023).
>
> [6] Scaling-up remote sensing segmentation dataset with segment anything model. arXiv preprint (2023).
>
> [7] Diffumask: Synthesizing images with pixel-level annotations for semantic segmentation using diffusion models. arXiv preprint (2023).
>
> [8] HandsOff: Labeled dataset generation with no additional human annotations. CVPR. 2023.
>
> [9] Datasetgan: Efficient labeled data factory with minimal human effort. CVPR. 2021.

---

> > ### Comment · Area_Chair_yWWh · 2023-08-18
> > **discussion**
> >
> > Dear Reviewer rde2,
> >
> > Thank you for being a reviewer for NeurIPS2023, your service is invaluable to the community!
> >
> > The authors have submitted their feedback.
> >
> > Could you check the rebuttal and other reviewers' comments and start a discussion with the authors and other reviewers?
> >
> > Regards,
> > Your AC

---

> > ### Comment · Reviewer_rde2 · 2023-08-19
> >
> > Thank the authors for the feedback.
> >
> > After reading the authors' feedback and other reviews, I have several remaining concerns and look forward to further responses.
> >
> > It can be expected and understood that with more real images, the improvement brought by synthetic data will gradually diminish. However, I think the 1.1% improvement of full real data is a little marginal, especially considering that we can indeed directly use other real unlabeled data sources to boost our target set, without requiring the synthesis stage. For example, we may use the COCO 118K images as unlabeled images, and use the model trained on Pascal to assign pseudo labels for COCO images in a semi-supervised manner, similar to the investigation in [1]. Unlabeled COCO images may boost the Pascal data similarly, or more significantly, because the COCO images are real images instead of relatively lower-quality synthetic images. Could the authors provide any discussions on this or provide some experiments?
> >
> > I agree with Reviewer DG4j that it is highly recommended to conduct thorough comparisons with semi-supervised works. I am looking forward to seeing the comparisons the authors are conducting. A piece of side advice is that the authors are more suggested to use better semi-supervised semantic segmentation methods than ReCo, such as [2], if the time allows.
> >
> > Another confusing thing is that there are only 10,582 training images on Pascal. The 11.5k real images the authors mentioned above seem to have included the 1,449 validation images. However, the validation images can not be used for training. Are there any misunderstanding here?
> >
> > [1] Zoph, Barret, et al. "Rethinking pre-training and self-training." NeurIPS 2020.
> >
> > [2] Yang, Lihe, et al. "Revisiting weak-to-strong consistency in semi-supervised semantic segmentation." CVPR 2023.

---

> > > ### Author Response · Authors · 2023-08-19
> > >
> > > Thank you for the reviewer's response！
> > >
> > > **Q1**: Using the COCO 118K images as unlabeled images, and use the model trained on Pascal to assign pseudo labels for COCO images in a semi-supervised manner, similar to the investigation in [1].
> > >
> > > **A1-1(experiment)**: We aim to provide experiments with relevant settings. We will utilize the model trained with mask2former on Pascal (84.3 mIoU) to generate pseudo-labels for COCO's 118K images. Subsequently, we will mix all the data and retrain the model. The relevant table is provided below, and we will update it accordingly.
> > >
> > > | |  10,582 images| 10,582 images, 40k synthetic images | 10,582 images, Unlabeled 118K (COCO)|10,582 images, Unlabeled 118K (COCO),40k synthetic images |
> > > |  ----  | ----  |----  |----  | ----  |
> > > | mIoU | 84.3 | 85.4 | 84.3 | 85.3 |
> > >
> > > Note: Due to time constraints, we are currently unable to facilitate the migration of [1] onto mask2former. Additionally, conducting large-scale experiments using the official [1] code (https://github.com/tensorflow/tpu/tree/master/models/official/detection/projects/self_training) is not feasible within a short timeframe. The code utilizes TensorFlow and involves unfamiliar training, configuration, and inference architecture. It also involves extensive Data Augmentation and self-training on COCO (240k images, combining labeled and unlabeled datasets), as well as ImageNet  (1.2M images).
> > >
> > > **A1-2(discussion).** Using large amounts of real unlabeled data sources to boost the target set is a well-explored area in the field, with numerous related works. However, there are also some intuitively foreseeable advantages of synthetic data:
> > >
> > > - **Sensitivity and accessibility**. In domains sensitive to data (such as faces or medical images), where obtaining real data might be contentious or unfeasible, synthetic data presents a viable solution. We provide corresponding performance comparisons for face data, specifically Celebi-Mask-HQ, in the rebuttal PDF (Table 5).
> > >
> > > - **Zero Shot(Open-Vocabulary) Segmentation.** Our work also supports the Zero Shot (Open-Vocabulary) Segmentation task. Please refer to Section 3.4 for the Open-Vocabulary Segmentation method, experimental results in Table 5 of the main paper, as well as Table 9 in the supplementary material. It seems that semi-supervised semantic segmentation methods are unable to support this.
> > >
> > > - **Six Tasks.** Our method can simultaneously support semantic segmentation, instance segmentation, depth estimation, pose estimation, long-tail segmentation, and zero-shot segmentation tasks. To our knowledge, there is no semi-supervised method that can accommodate such a diverse range of tasks. Moreover, some semi-supervised segmentation techniques that work in one area may not be applicable to others. For instance, data augmentation methods like classmix [3] may not be suitable for depth estimation when there's no clear distinction between foreground and background in the depth information.
> > >
> > > - **Strong automated customization**.  Synthetic data offers strong potential for automated customization, effectively combining with language models (GPT4) to innovate datasets with a vast array of granular content. This can be observed in Fig. 4, Fig. 5 of the paper, and Fig. 1, 2, and 4 of the supplementary material. This scenario can also be observed in Fig. 6 of [2], where can synthesize more fine-grained masks. For example, generating synthetic images for 200 different bird classes based on text-guided, using unlabeled real images seems to be a task that cannot be effectively handled.
> > >
> > > **Conclusion.** Even in comparison to the setting of semi-supervised semantic segmentation, it seems that our work also provides some irreplaceable inspiration and contribution to the community.
> > >
> > > **Q2**: Better semi-supervised semantic segmentation methods than ReCo, such as [4].
> > > **A2**: Thank you for the constructive feedback. We will include this method in the comparison for semi-supervised semantic segmentation as well. We will provide updates here as there are further developments.
> > >
> > > **Q3**: Confusion for 10,582 training images or 11.5k real images.
> > > **A3**:  Sorry to confuse you, this was a wording mistake. Similar to all previous works, we only used 10,582 training images and did not use the 1,449 validation images. We will correct this, and we appreciate your reminder.
> > >
> > > [1] Zoph, Barret, et al. "Rethinking pre-training and self-training." NeurIPS 2020.
> > >
> > > [2] Diffumask: Synthesizing images with pixel-level annotations for semantic segmentation using diffusion models. arXiv (2023).
> > >
> > > [3] Classmix: Segmentation-based data augmentation for semi-supervised learning. WACV, 2021.
> > >
> > > [4] Yang, Lihe, et al. "Revisiting weak-to-strong consistency in semi-supervised semantic segmentation." CVPR 2023.

---

> > > > ### Author Response · Authors · 2023-08-20
> > > >
> > > > **Remaining Concern 1: Unlabeled real image, synthetic data**. We provide the corresponding experiments as follows:
> > > > | |  10,582 images| 10,582 images, 40k synthetic images | 10,582 images, Unlabeled 118K (COCO)|10,582 images, Unlabeled 118K (COCO),40k synthetic images |
> > > > |  ----  | ----  |----  |----  | ----  |
> > > > | mIoU | 84.3 | 85.4 | 84.3 | 85.3 |
> > > >
> > > > Firstly, before delving into the analysis of the relationship between unlabeled real images and synthetic data, let's start by examining `what exactly contributes to better performance from a data perspective`:
> > > >
> > > > **Condition 1: Image Quality.** Image quality encompasses factors such as image resolution, complexity of the scene, and diversity of images. When the annotation quality is very high, higher image quality and greater quantity naturally lead to further performance improvements. This is a reasonable expectation.
> > > >
> > > > **Condition 2: Annotation Quality.** When the quality and quantity of the data remain constant, improving annotation quality and accuracy can also lead to significant improvements.
> > > >
> > > > Here, we also provide some of our analyses for unlabeled real image and synthetic data as follows:
> > > >
> > > > - **High-quality real images result in low-quality pseudo labels**. As the reviewer pointed out, COCO's real images have higher image quality than synthetic ones. However, performance isn't just about image quality—it's influenced by both image and pseudo label quality. `Complex scenes may lead to low-quality pseudo masks`, which could have a more significant negative impact than the benefits gained from images. This is also a challenge that semi-supervised segmentation aims to address: Improving the quality of pseudo masks or mitigating their negative effects.
> > > >
> > > > - **Low-quality synthetic images result in high-quality pseudo labels**. Similarly, lower-quality synthetic images can yield more accurate masks. When the synthetic images are less complex, their mask accuracy improves. However, the overall image quality must also be enhanced for ultimate performance gains. `For synthetic data, we can enhance images through various data augmentation techniques, such as Splicing, Gaussian Blur, Occlusion, and Perspective Transform[1]`.
> > > >
> > > > - **Complementary, not replacing**.  In fact, we do not need to choose between unlabeled real images and synthetic data, as they can also be used simultaneously. When both can provide benefits, mixed training appears to be the best choice.
> > > >
> > > > **Conclusion.** For unlabeled data and synthetic data, it's difficult to provide a definitive answer about which one performs better. However, we know that these two are `complementary rather than in competition`. Synthetic data can further enhance the performance of both semi-supervised and fully supervised approaches. Additionally, synthetic data possesses some `advantages that real data lacks` (as mentioned in the previous response). Therefore, there is no doubt that our work provides irreplaceable inspiration and contribution to the community.
> > > >
> > > > **Remaining Concern 2: Comparisons with semi-supervised works.** We are currently working diligently to generate 40k synthetic images with 200 real images and train DeepLab V3. We aim to provide the result by tomorrow noon.
> > > >
> > > > | method | baseline | backbone| labeled real image | unlabeled real image | synthetic image | mIoU |
> > > > |  ----  | ----  |----  |----  | ----  |----  | ----  |
> > > > | ReCo +ClassMix| DeepLabv3 | R101 |60 images| 11.5k-60 | 0 | `53.31` |
> > > > | ReCo +ClassMix|DeepLabv3 | R101 |  200 images | 11.5k-200 | 0| `69.81` |
> > > > | ours | DeepLabv3 | R101 |60 images| 0 | 40k | 57.64 |
> > > > | ours | DeepLabv3 | R101 |200 images| 0 | 40k  | |
> > > > | ours | Mask2former| R50 |100 images | 0 | 40k  | 66.10|
> > > > | ours | Mask2former| Swin-B|100 images | 0 | 40k  | 78.50|
> > > >
> > > > **Remaining Concern 3: Comparisons with [2].** Comparing the settings of [1] and [2] (training set size and split), it appears that they are not uniform. We apologize for not being able to provide the experimental comparison within `two` days. However, we would like to reiterate that our approach is `complementary` to most semi-supervised and supervised methods. In this context, we believe that the contribution and significance of our approach may not necessarily hinge on outperforming all semi-supervised methods, especially when they utilize an extensive amount of real images. Furthermore, our method possesses `four significant advantages` not present in semi-supervised methods, as mentioned in the previous response.
> > > >
> > > >
> > > >
> > > > Finally, we sincerely appreciate the positive feedback and discussions provided by Reviewer #rde2, thanks !
> > > >
> > > > [1] Diffumask: Synthesizing images with pixel-level annotations for semantic segmentation using diffusion models. arXiv (2023).
> > > >
> > > > [2] Yang, Lihe, et al. "Revisiting weak-to-strong consistency in semi-supervised semantic segmentation." CVPR 2023.
> > > >
> > > > [3] Bootstrapping semantic segmentation with regional contrast. arXiv preprint arXiv:2104.04465 (2021).

---

> > > > > ### Comment · Reviewer_rde2 · 2023-08-21
> > > > >
> > > > > Thank the author very much for the further results.
> > > > >
> > > > > I agree with most of the authors' claims about the differences between real and synthetic images. However, as for the second difference "low-quality synthetic images result in high-quality pseudo labels",  I politely disagree with it. Even though real unlabeled images are more challenging to assign pseudo labels, they are much more informative to learn than synthetic images (also supported by results in Response 3  above, 100 real images: 65.3 *vs.* 100 synthetic images: 40.1). For the hard-to-predict regions, we can easily skip them during learning by setting a pixel-wise threshold for them, which is a common practice in semi-supervised learning. I believe the reported COCO results above can be much higher when applying a stronger semi-supervised framework and set an ideal threshold.
> > > > >
> > > > > The contribution of this work seems to only lie in replacing the real unlabeled images with synthetic unlabeled images, which can be deemed as a side work of semi-supervised learning paradigms. Moreover, there is no strong evidence to prove that synthetic images can play a superior role to real unlabeled images. There is also a severe domain gap issue when using these generative models for some rare domains, such as medical images and remote sensing images.
> > > > >
> > > > > Based on the above considerations, I would change my rating to 4: Borderline Reject.

---

> > > > > > ### Author Response · Authors · 2023-08-21
> > > > > >
> > > > > > Dear Reviewer rde2,
> > > > > >
> > > > > > We believe there might be some misunderstandings there:
> > > > > >
> > > > > > **Q**: Even though real unlabeled images are more challenging to assign pseudo labels, they are much more informative to learn than synthetic images (also supported by results in Response 3 above, 100 real images: 65.3 vs. 100 synthetic images: 40.1).
> > > > > >
> > > > > > **A**: First, the ground truth for the 100 real images is manually annotated, resulting in a mIoU of 65.3. For the 100 synthetic images, a pseudo label is utilized, resulting in a mIoU of 40.1. Directly comparing the both may appear unfair. A more reasonable approach would involve using 100 real image with pseudo labels generated by a model trained on 100 real images.
> > > > > > Otherwise, a comparison needs to be made against 100 real images + 100 synthetic images (mIoU of 67.9), even 100 real images + 40k synthetic images (mIoU of 78.5).
> > > > > >
> > > > > > **Q**: The contribution of this work seems to only lie in replacing the real unlabeled images with synthetic unlabeled images, which can be deemed as a side work of semi-supervised learning paradigms.
> > > > > >
> > > > > > **A**: Firstly, as we have explained above, synthetic data can serve as a complement to real data, not a replacement. Secondly, the reviewer should take into account the `four advantages` we mentioned in our previous response, which real data lacks (semi-supervised learning): Sensitivity and accessibility. Zero Shot (Open-Vocabulary) Segmentation. Six Tasks. Strong automated customization. If possible, before reducing the score rating, it is better to provide a response accordingly.
> > > > > >
> > > > > > **Q**:  Moreover, there is no strong evidence to prove that synthetic images can play a superior role to real unlabeled images.
> > > > > >
> > > > > > **A**:  Under the same setting, with 60 images, our approach can outperform the semi-supervised algorithm (57.64 vs. 53.31). Furthermore, exploring the training with a combination of semi-supervised and synthetic datasets is also a valuable approach to consider, without necessarily choosing one over the other.
> > > > > >
> > > > > >
> > > > > > Regardless, the authors appreciate the constructive discussion provided by reviewer rde2.

---

### Official Review · Reviewer_21oc · 2023-07-10

**Soundness:** 2 fair
**Presentation:** 3 good
**Contribution:** 2 fair
**Rating:** 5
**Confidence:** 5

**Summary:**

Large-scale datasets with quality annotations are often costly and time-consuming to collect and annotate. Since for most applications, large amounts of annotated data are required by deep neural networks, the authors propose a framework for text-guided synthetic data generation - called DatasetDM - which can produce an infinite number of images and their corresponding annotations, such as semantic segmentation masks or depth maps. Their method builds upon a pretrained diffusion model (specifically, Stable Diffusion) which is extended to include a simple decoder tasked with translating the latent space of the diffusion model’s time-conditional UNet to what the authors call “perception annotations”. The method is evaluated on several downstream tasks: semantic segmentation, instance segmentation, depth map prediction, human pose estimation, and zero-shot semantic segmentation. For each proposed downstream task, the authors use less than 1% of the training dataset to train the perception decoder on multi-scale text-to-image representations, and demonstrate competitive or improved performance over baselines.

**Strengths:**

- The paper attacks a problem relevant to the field: how to efficiently generate pairs of synthetic images and corresponding labels for a variety of core tasks within computer vision, with a framework that requires only minor modifications in architecture or representation depending on the task.

- The framework seems adaptable to several tasks of interest in computer vision: depth map prediction, semantic segmentation, instance segmentation, open-vocabulary version of the latter, and pose estimation.

- The paper is clearly written and easy to follow.

- Results across several downstream tasks (specifically semantic segmentation and instance segmentation) show impressive improvement over baselines.

- Interesting observation that training on synthetic and real data in the ablation studies section helps raise mIOU (Table 4) by a nontrivial amount.


**Weaknesses:**

The evaluation methodology and experimentation proposed in the paper is generally sound, but there are a few notes I'd like to make on the choice of baselines and dataset selection, which would strengthen the contribution claims of the paper.

Choice of baselines / experiments

- The authors cite the recently published work HandsOff [1], which proposes a very similar dataset generation paradigm – although utilizing the latent space of a pretrained StyleGAN model instead of Stable Diffusion, and using GAN inversion instead of diffusion inversion to create a hypercolumn representation per pixel (that is then fed through a simple label generator to produce similar “perception labels”: semantic segmentation masks, depth maps, human pose keypoint prediction, etc.).

- HandsOff uses fewer images for all downstream tasks (16 or 50 depending on the dataset and task), and also uses the same datasets for evaluation on downstream task performance as other cited works like DatasetGAN [2], BigDatasetGAN [3], and EditGAN [4]. In addition, [1], [2] and [4]  also share the same class consolidation in each dataset, making comparison easier. A comparison on similar datasets and similar class structure to these papers (eg. CelebA-Mask-HQ [6] or DeepFashion-MM [9]) with a similar number of training images (instead of the variable “1% of the training dataset” the authors use, which they say averages to around 100 training images) could be added in the performance comparison to establish more firmly a SOTA claim across the proposed tasks.

- I would like to see a comparison against DatasetDDPM [7] – a diffusion-model-based synthetic dataset generation method akin to the proposed approach and concurrent work – to give the reader a more comprehensive understanding of model performance. However, as per the supplementary material of [1], HandsOff actually establishes SOTA performance over all of [2], [4], and [7] with fewer labelled images, thus making a comparison against DatasetDDPM redundant. This could challenge the claim the authors make on L42-44: “Due to the limitations of the representation ability of previous GAN models, the quality of the synthesized data is often dissatisfactory, leading to an inferior performance on downstream tasks”. I see a comparison against HandsOff in the supplementary on Cityscapes semantic segmentation, where the proposed model does indeed outperform, but the same is missing for the pose estimation evaluation on DeepFashion-MM [9]. The best result presented is mIOU of 45.1 with 100 labelled images in 24 classes and with the Swin-B backbone, while HandsOff presents best mIOU of 68.4 with 50 labelled images in 8 classes and with the ResNet151 backone.

- In summary, what would be beneficial to add to the baselines section is a fair comparison on pose estimation on DeepFashion-MM [9] against HandsOff, and a comparison on semantic segmentation on CelebA-Mask-HQ [6] (where HandsOff outperforms DatasetDDPM, another diffusion-based data generation framework).

Missing references

- EditGAN [4] and DatasetDDPM [7] should be added to the list of citations.

Minor

- The authors refer to results on the Cityscapes dataset typically used for semantic segmentation evaluation (L242), and outline an example of a generated prompt inputted to Stable Diffusion to generate images akin to the urban driving scenes (L226+). I did not see a reference to the supplementary for a table of the results; I also think mIOU improvement is significant and could be added to Table 3 in the main paper.

- L60: possible typo: “... with minimal human-labeled…”

- L241 states: “To comprehensively evaluate the generative image of DatasetDM, we conduct seven groups of experiments for the supported six downstream tasks”, but L261 says “Tab. 1 provides a basic comparison of the selected four downstream tasks.” I also found it confusing from the introduction and abstract to form an expectation as to which downstream tasks the authors evaluated on, and whether they achieve SOTA performance on all or some. It would be helpful to enumerate specific tasks with SOTA results in the abstract and in the claims section of the introduction.

References

[1] A. Xu, M. I. Vasileva, A. Dave, and A. Seshadri. Handsoff: Labeled dataset generation with no additional human annotations. In Proceedings of the IEEE/CVF Conference on Computer Vision and Pattern Recognition (CVPR), 2023.

[2] Y. Zhang, H. Ling, J. Gao, K. Yin, J.-F. Lafleche, A. Barriuso, A. Torralba, and S. Fidler. Datasetgan: Efficient labeled data factory with minimal human effort. In Proceedings of the IEEE/CVF Conference on Computer Vision and Pattern Recognition (CVPR), 2021.

[3] ] D. Li, H. Ling, S. W. Kim, K. Kreis, S. Fidler, and A. Torralba. Bigdatasetgan: Synthesizing imagenet with pixel-wise annotations. In Proceedings of the IEEE/CVF Conference on Computer Vision and Pattern Recognition (CVPR), 2022.

[4] Huan Ling, Karsten Kreis, Daiqing Li, Seung Wook Kim, Antonio Torralba, and Sanja Fidler. EditGAN: HighPrecision Semantic Image Editing. In Advances in Neural Information Processing Systems (NeurIPS), 2021.

[5] Marius Cordts, Mohamed Omran, Sebastian Ramos, Timo Rehfeld, Markus Enzweiler, Rodrigo Benenson, Uwe Franke, Stefan Roth, and Bernt Schiele. The Cityscapes Dataset for Semantic Urban Scene Understanding. In Proceedings of the IEEE/CVF Conference on Computer Vision and Pattern Recognition (CVPR), 2016.

[6] Cheng-Han Lee, Ziwei Liu, Lingyun Wu, and Ping Luo. MaskGAN: Towards Diverse and Interactive Facial Image Manipulation. In Proceedings of the IEEE/CVF Conference on Computer Vision and Pattern Recognition (CVPR), 2020.

[7] Dmitry Baranchuk, Andrey Voynov, Ivan Rubachev, Valentin Khrulkov, and Artem Babenko. Label-Efficient Semantic Segmentation with Diffusion Models. In International Conference on Learning Representations (ICLR), 2022.

[8] B. Cheng, I. Misra, A. G. Schwing, A. Kirillov, and R. Girdhar. Masked-attention mask transformer for universal image segmentation. In Proceedings of the IEEE/CVF Conference on Computer Vision and Pattern Recognition (CVPR), 2022.

[9] Ziwei Liu, Ping Luo, Shi Qiu, Xiaogang Wang, and Xiaoou Tang. DeepFashion: Powering Robust Clothes Recognition and Retrieval with Rich Annotations. In Proceedings of the IEEE/CVF Conference on Computer Vision and Pattern Recognition (CVPR), June 2016.

**Questions:**

- Please see a list of suggested baseline comparisons in the previous section which I believe would better support the paper's claims.

- How were the L(~100) prompts generated per dataset created? What was the process of quality control on the generated prompts to ensure the desired diversity in class vocabulary/labels in each respective dataset and task?

- In Table 4, the difference in performance with an increased number of training images seems to demonstrate an increasing trend. When does performance peak?

- Prompt length seems to be of crucial contribution to performance improvements - how feasible is it to provide high-quality, diverse prompts in practice, for any dataset and task? Will the prompt candidates need a human-in-the-loop quality control procedure?

**Limitations:**

Nothing of note with regard to potential negative societal impact of the proposed work.

---

> ### Author Rebuttal · Authors · 2023-08-06
>
> Thanks to the reviewers for the valuable comments.
>
> **Weakness 1-1**: Comparison against Handsoff at the **method** levels.
> **R1**: We provide a comparison as follows:
>
> - **Task.** HandsOff designs several CNN layers for the decoder, which cannot handle tasks such as instance segmentation and open-vocabulary segmentation. In contrast, we design a transformer-based approach to accommodate these tasks, emphasizing that instance segmentation and open-vocabulary segmentation are more challenging compared to semantic segmentation.
>
> - **Architecture.** Our method and HandsOff are completely different for the architecture, even in the choice of feature extraction models (GAN and Diffusion). Moreover, the decoders have fundamental differences, with one using several CNN layers and the other one utilizing a transformer-based approach (see Fig.2 and Fig. 3).
>
> **Weakness 1-2**: Comparison against Handsoff at the **experiment** level.
> **R2**: In fact, our method outperforms HandsOff by a large margin as follows:
>
> - **Semantic Segmentation on DeepFashion-MM.** See Tab. 1 of the Rebuttal PDF, under the same setting, using fewer synthetic data (40k v.s 100k+), we can achieve an improvement of nearly `3%`.
>
> - **Semantic Segmentation on Cityscapes.** See Tab. 2 of the Rebuttal PDF (Information from Tab. 3 of Supplementary Material). With 9 real images, our method surpasses Handsoff using more data (50 images), achieving `17.1%` mIoU improvement for the 8-class split. HandsOff did not provide the corresponding performance for the 19-class split.
>
> **Weakness 1-3**: Handsoff challenges the claim the authors make on L42-44.
> **R3**: In fact, this is an open and unresolved issue (GAN or diffusion, who is better), and we would like to share my perspective:
>
> - **GAN v.s. Diffusion.** Currently, most of the SOTA models are diffusion-based, including well-known ones such as Stable Diffusion by Stability AI [3], Imagen by Google [4], and DALL·E 2 by OpenAI [5]. Some researchs[6] has attempted to compare Diffusion and GAN and determine which is better for generation, but there seems to be no definitive conclusion yet.
>
> - **Experiment.** The comparison between Handsoff and DatasetDDPM cannot conclusively determine whether GAN are superior to diffusion model. Firstly, they only compared one dataset, CelebAMask-HQ (Tab. 1 of Handsoff), which contains only human face data. Secondly, both StyleGAN used in Handsoff and DDPM[2] used in DatasetDDPM were proposed in 2020. Newer models like Stable Diffusion have achieved significantly better results.
>
> - **Correction.** Our claim may not be the most appropriate, and we will revise it to 'Due to the limitations of the representation ability of early (up to 2020) GAN models.'
>
> **Weakness 2**: Comparison against DatasetDDPM [1]
> **R4**: We provide the following comparisons between similarities and differences:
>
> - **Different Task**. DatasetDDPM focuses solely on semantic segmentation. In contrast, we propose a versatile dataset generation model that can simultaneously handle `six` tasks, including instance segmentation, and human pose estimation(see Fig. 1, Tab. 5).
>
> - **Different Architecture**. DatasetDDPM utilizes five features extracted from the UNet (DDPM[2]), while we use four features (L173) from the UNet of Stable Diffusion. Additionally, the perception decoder is entirely different. DatasetDDPM employs several CNN layers to perform a multi-class task for one mask, which cannot handle tasks such as instance segmentation, and open-vocabulary segmentation. In contrast, we use a unified transformer-based decoder (see Fig. 3 and Fig. 4) that can simultaneously handle six downstream tasks.
>
> - **Diffusion-based (similarity)**. Although the diffusion models we use are different (DDPM and Stable Diffusion), they both belong to the diffusion models.
>
> **Summary**: Our method and DatasetDDPM[1] have fundamental differences in tasks and core contributions (perception decoder architecture).
>
> **Weakness 3**: Experiment for Celebi-Mask-HQ.
> **R5**. We presented experiments in the rebuttal PDF (Tab. 5). We achieve an improvement of nearly 1.5%.
>
> **Weakness 4**: Missing references and Minor.
> **R6**: Thank you for the valuable suggestion. We will cite and discuss relevant works in the latest version and make minor corrections.
>
> **Weakness 5**: I did not see a table of the Cityscapes.
> **R7**: Supplementary Material Table 3 and Section 2.4.
>
> **Weakness 6**: Confusion.
> **R8**: A total of six downstream tasks, Tab. 1 presents the basic four of them, and more are available in Tab. 5 and the supplementary material.
>
> **Q1**:Baseline comparisons
> **R9**: See R1 - R5.
>
> **Q2**:How were the prompts generated?
> **R10**: In the supplementary materials, we provide detailed prompt examples for each dataset and task, as well as the generation process in Tab. 11 and Fig. 4.
>
> **Q3**:In Table 4, When does performance peak?
> **R11**: More experimental results can be found in Tab. 3 (increasing real data) and Tab. 4 (increasing synthetic data) of the rebuttal PDF. In brief, when we continuously increase the real data, the performance keeps improving (no peak). When we continue to increase the synthetic data, as the data exceeds 100k, further data increase has little impact on result improvement.
>
> **Q4**:Will the prompt candidates need a human-in-the-loop quality control procedure?
> **R12**: As shown in Fig. 4 of the supplementary materials, humans only need to provide a few `simple prompts` without any other manual intervention. These prompts can be completed in just `a few minutes`.
>
> [1]  Label-Efficient Semantic Segmentation with Diffusion Models. ICLR 2022
>
> [2] Denoising diffusion probabilistic models. NeurIPS 2020
>
> [3] High-resolution image synthesis with latent diffusion models.CVPR. 2022
>
> [4] Photorealistic text-to-image diffusion models with deep language understanding.NeurIPS 2022
>
> [5] Zero-shot text-to-image generation. ICML, 2021.
>
> [6] Diffusion models beat gans on image synthesis. NeurIPS 2021

---

> > ### Comment · Area_Chair_yWWh · 2023-08-18
> > **discussion**
> >
> > Dear Reviewer 21oc,
> >
> > Thank you for being a reviewer for NeurIPS2023, your service is invaluable to the community!
> >
> > The authors have submitted their feedback.
> >
> > Could you check the rebuttal and other reviewers' comments and start a discussion with the authors and other reviewers?
> >
> > Regards,
> > Your AC

---

### Author Rebuttal · Authors · 2023-08-08

Thanks to all reviewers and ACs for the valuable comments and suggestions. We have summarized and addressed the main concerns raised by each reviewer as follows:

**To Reviewer 21oc.** Our work differs from Handsoff and DatasetDDPM primarily in terms of uniqueness and contribution (lines 69 - 85). The key distinctions lie in the fact that we cater to `a wider range of tasks`, unlike Handsoff and DatasetDDPM, which cannot handle instance and open-vocabulary segmentation. Additionally, there are fundamental `differences in perception decoder`. While we utilize a unified transformer decoder, the other two approaches rely on CNN-based methods. Besides, our method `outperforms` HandsOff by a large margin (see Tab. 1, Tab. 2 and Tab. 5 of the Rebuttal PDF).

**To Reviewer rde2.** Our synthesized data can boost an already well-established baseline with all available real training data (`11.5k`) to achieve a `1.1%` increase in mIoU on VOC 2012. Furthermore, we would like to emphasize that the primary motivation of this paper (lines 23-33) is to decrease annotation costs and offer valuable insights to the community, highlighting the strong potential of synthesized data.

**To Reviewer PHiv.** Thank you for your constructive feedback for the number of training samples. However, in fact, training with 400 real images compared to 80k synthetic images is `fair`, as their annotation costs are consistent. The motivation of the paper is to decrease the cost of data (refer to L 22-31). And our approach `aligns` completely with prior studies [1][2][3], all of which investigate utilizing a small quantity of real data to acquire a substantial amount of effective synthetic data at a low cost. Furthermore, in Semantic Segmentation on VOC 2012, using the same quantity of synthetic data (100 images) as the real data, we attain a `2.7%` increase in mIoU, and with 400 synthetic data, there's a `10.9%` enhancement. Even with the use of 11.5k real data (full training images, more realistic baselines), our synthetic data still yield a `1.1%` mIoU improvement.

**To Reviewer DG4j and aPc6.** Firstly, we want to emphasize that our work and VPD (unpublished) are concurrent works. Secondly, our work and theirs are fundamentally different in terms of `tasks`, core `contributions`, and `architecture`. Here are some comparisons:

- **Different Task**. Our work is a synthetic data augmentation for perception tasks, while VPD is a diffusion-based perception algorithm. In addition, we support more tasks than VPD, including Pose Estimation, Fashion Segmentation, and Open-Vocabulary Segmentation, which are not merely simple extensions but require corresponding improvements in the perception decoder structure.

- **Different Architecture**. The most crucial design component in VPD or our work is the perception decoder, and in this aspect, we are completely different. We design a transformer-based approach to model multiple tasks (Fig. 3 of paper), while VPD employs existing decoder models (CNN-based), such as FPN for semantic segmentation. The task-specific decoder of VPD utilizes multiple architectures, and can not support instance segmentation and open-vocabulary segmentation.

- **Training strategy**. We explore few-shot tuning by generating more synthetic data for auxiliary training. VPD, on the other hand, focuses on training with the entire dataset.

- **Summary**. We have numerous differences with VPD for core contribution (L69-85 of our paper), with the most significant being the dissimilarity in decoder structure and the supported tasks. VPD can not support for instance segmentation, pose estimation, zero-shot segmentation, and long-tail segmentation.

**To all Reviewers.** We sincerely thank and appreciate the diligent efforts and valuable insights from all the reviewers for our paper. The authors look forward to further responses and discussions with you in the author-reviewer discussion period.


Best regards,

Authors of paper 5020.


[1] Datasetgan: Efficient labeled data factory with minimal human effort. CVPR. 2021.

[2] BigDatasetGAN: Synthesizing ImageNet with pixel-wise annotations. CVPR. 2022.

[3] HandsOff: Labeled dataset generation with no additional human annotations. CVPR. 2023.

---

### Decision · Program_Chairs · 2023-09-21

**Decision:**

Accept (poster)

**Comment:**

This paper received three positive reviews and two negative reviews. Since this paper is borderline, the AC did a detailed review of the paper, the authors' feedback, and the reviewers' comments and discussions. One of the concerns of the reviewers who gave a negative score is the difference from the VPD; the VPD was indeed submitted to arXiv more than two months before NeurIPS, but given that the upload date is March 3, it can be considered almost concurrent work. The authors have also carefully explained the differences from the VPD in their feedback, and have performed many comparison experiments in the limited time available to demonstrate the differences. The proposed method loses when trained with only the same number of synthetic data as real data, but this is not a problem since the number of synthetic data can be easily increased.

In conclusion, the ACs consider this paper to be accepted, but strongly recommend that the reviewers' comments be addressed in the camera-ready paper, for example by describing the differences from VPD and including additional experiments.